# Heterogeneity in the *Drosophila* gustatory receptor complexes that detect aversive compounds

Ha Yeon Sung [1], Yong Taek Jeong [2], Ji Yeon Lim [3], Hyeyon Kim [2], Soo Min Oh [2], Sun Wook Hwang [3], Jae Young Kwon [1] & Seok Jun Moon [2]

Animals must detect aversive compounds to survive. Bitter taste neurons express heterogeneous combinations of bitter receptors that diversify their response profiles, but this remains poorly understood. Here we describe groups of taste neurons in *Drosophila* that detect the same bitter compounds using unique combinations of gustatory receptors (GRs). These distinct complexes also confer responsiveness to non-overlapping sets of additional compounds. While either GR32a/GR59c/GR66a or GR22e/GR32a/GR66a heteromultimers are sufficient for lobeline, berberine, and denatonium detection, only GR22e/GR32a/GR66a responds to strychnine. Thus, despite minimal sequence-similarity, *Gr22e* and *Gr59c* show considerable but incomplete functional overlap. Since the gain- or loss-of-function of *Gr22e* or *Gr59c* alters bitter taste response profiles, we conclude a taste neuron's specific combination of *Grs* determines its response profile. We suspect the heterogeneity of *Gr* expression in *Drosophila* taste neurons diversifies bitter compound detection, improving animal fitness under changing environmental conditions that present a variety of aversive compounds.

[1] Department of Biological Sciences, Sungkyunkwan University, Suwon, Gyeonggi-do 16419, Korea. [2] Department of Oral Biology, BK21 PLUS Project, Yonsei University College of Dentistry, Yonsei-ro 50-1, Seodaemun-gu, Seoul 03722, Korea. [3] Department of Biomedical Sciences and Department of Physiology, Korea University College of Medicine, Seoul 02841, Korea. Ha Yeon Sung, Yong Taek Jeong and Ji Yeon Lim contributed equally to this work. Correspondence and requests for materials should be addressed to J.Y.K. (email: jykwon@skku.edu) or to S.J.M. (email: sjmoon@yuhs.ac)

Animals use their sense of taste mainly for evaluating food quality. Animals detect potentially harmful compounds in their food to avoid ingesting toxic chemicals. Many toxic compounds taste bitter and induce aversive behaviors. Since many plants defend themselves against herbivory by producing a diversity of bitter compounds, many herbivorous insects have evolved gustatory systems capable of detecting large numbers of bitter compounds.

Higher concentrations of these bitter compounds elicit more robust behavioral aversion, suggesting bitter-responsive cells discriminate concentration[1]. It remains unclear, however, whether the bitter taste modality perceives a generic "bitterness" leading to a generic behavioral aversion or whether the system shows more specific molecular discernment permitting more subtle, complex behavioral responses. The bitter-responsive cells of several species reportedly express heterogeneous groups of bitter receptors[2, 3] and they respond to distinct bitter compounds[3–5]. Still, the consequences of this heterogeneity of bitter receptor expression have not been fully explored at the molecular and cellular levels.

*Drosophila* have taste organs all over the body including the labellum, legs, pharynx, anterior wing margin, and even the female genitalia[6, 7]. Of these locations, the *Drosophila* labellum is an attractive place to study bitter taste receptor heterogeneity. The bitter-responsive taste neurons of the labellum are easily accessible for electrophysiological analyses, and we have many genetic reagents that permit precise control over bitter receptor expression in the bitter-responsive taste neurons. Each half-labellum contains 31 taste sensilla classified by their relative length and location into Large (L)-type, Intermediate (I)-type, and Small (S)-type[8]. According to electrophysiological analyses of these 31 taste bristles, all three types of sensilla show distinct response profiles to bitter compounds[3]. After mapping these response profiles to the various morphologic classes, it became clear that the labellar taste sensilla should actually be divided into five classes (Fig. 1a)[3]. L-type and S-c sensilla show almost no response to bitter compounds. S-a and S-b sensilla are broadly tuned, with S-b sensilla showing robust responses to most bitter compounds. I-a and I-b sensilla are narrowly tuned, with complementary response profiles to bitter compounds[3].

Several classes of chemosensory receptors have been suggested as taste receptors for the detection of aversive chemicals[9–11], but most bitter compound detection requires members of the gustatory receptor (GR) family. The 60 *Gr* genes in the fly genome encode 68 proteins by alternative splicing[12, 13]. The bitter gustatory receptor neurons (GRNs) of each sensilla class (i.e., S-a, S-b, I-a, and I-b) express distinct *Gr* subsets[3]. For example, while bitter GRNs in S-a sensilla co-express 29 bitter *Grs*, bitter GRNs in I-a sensilla express a combination of 6 bitter *Grs*.

The role the *Grs* play in taste has been extensively studied by in vivo loss-of-function[14–20], but those experiments have limitations. Most of those studies focused on flies carrying mutations in individual *Grs*, measuring the bitter response profiles of very limited numbers of sensilla. Since bitter GRN response profile and *Gr* expression profile are different depending on sensilla type and since multiple independent GRs are required for bitter compound detection[20], it seems reasonable to assume the function of a given *Gr* depends on the *Grs* with which it is co-expressed. Indeed, in vivo misexpression of an individual *Gr* in different bitter GRNs induces differential effects, likely due to the other *Grs* they express[21]. Still, the precise molecular and cellular role *Grs* play in the detection of aversive chemicals remains unclear.

In this study, we attempt to parse the function of the bitter *Grs* in *Drosophila* in various molecular and cellular contexts. We found that while loss of either *Gr32a* or *Gr66a* abolishes sensitivity to lobeline (LOB), berberine (BER), and denatonium (DEN) in all labellar sensilla types, mutation of either *Gr22e* or *Gr59c* selectively impairs the detection of these chemicals in the S-b or I-a sensilla, respectively. Misexpression of either GR32a/ GR59c/GR66a or GR22e/GR32a/GR66a confers sensitivity to LOB, BER, and DEN on sweet GRNs and *Drosophila* S2 cells. We also found *Gr22e* expression in *Gr59c* mutant GRNs and *Gr59c* expression in *Gr22e* mutant GRNs rescues their detection of LOB, BER, and DEN. This suggests these two *Grs* are functionally redundant except for the additional detection of strychnine (STR) by the GR22e/GR32a/GR66a complex. Overexpression or misexpression of either *Gr22e* or *Gr59c* confers hypersensitivity or novel responsiveness to their respective agonists. The *Gr22e*, *Gr59c* double mutant shows a more severe defect in LOB, BER, or DEN avoidance than either the *Gr22e* or *Gr59c* single mutant. This may reflect an underlying potential for the graded modulation of repulsion to aversive chemicals. We have found that the detection of the same bitter compounds can be accomplished in different bitter GRNs by distinct bitter GR complexes. Since it is the specific combination of GRs expressed by each bitter GRN that determines its response profile, we propose the heterogeneity of bitter *Gr* expression across bitter GRNs diversifies bitter coding and broadens the behavioral repertoire with which insects respond to their chemical environment.

## Results

**Minimal receptors for bitter compound detection.** Labellar sensilla fall into five distinct functional classes depending on their electrophysiological response profiles (Fig. 1a)[3]. Of these, the I-a sensilla respond specifically to the aversive chemicals LOB, BER, and DEN (Fig. 1b). The bitter GRNs in I-a sensilla express six *Grs*: *Gr32a*, *Gr33a*, *Gr39a.a*, *Gr59c*, *Gr66a*, and *Gr89a* (Fig. 1b). We, therefore, asked whether these six GRs are sufficient to make functional LOB, BER, and DEN receptors, and if so, what the minimal GR components for LOB, BER, and DEN detection are. Since L-type sensilla do not respond to aversive chemicals[8], they are a convenient cell type in which to measure the responsiveness to aversive compounds conferred by misexpression of bitter *Gr* candidates.

To determine whether the combination of *Gr32a*, *Gr33a*, *Gr39a.a*, *Gr59c*, *Gr66a*, and *Gr89a* is sufficient for LOB, BER, and DEN detection, we misexpressed all six *Grs* in the sweet GRNs of L-type sensilla using *Gr64f-GAL4* (Fig. 1b)[22] and confirmed the misexpression of each *Gr* in the sweet GRNs by quantitative PCR (Supplementary Table 1). Importantly, control L-type sensilla respond only to sucrose, not to aversive chemicals (Fig. 1c, d). Although misexpression of all six *Grs* in L-type sensilla has no effect on sucrose responses (Fig. 1d), it confers robust responsiveness to LOB, BER, and DEN. Consistent with the response profile of I-a sensilla, misexpression of all six *Grs* in L-type sensilla does not confer responsiveness to caffeine (CAF), STR, sucrose octaacetate (SOA), or umbelliferone (UMB) (Fig. 1d). In further validation of these results, we found L-type sweet GRNs co-expressing all six *Grs* respond to LOB, BER, and DEN in a dose-dependent manner (Supplementary Fig. 1a).

We next sought to identify which *Grs* are indispensable in the detection of LOB, BER, and DEN. To do so, we misexpressed in the sweet GRNs of L-type sensilla six groups of five *Grs* at a time, omitting one of the original six *Grs* per group. We found the absence of *Gr32a*, *Gr59c*, or *Gr66a* prevents ectopic responses to LOB, BER, and DEN, suggesting these three *Grs* are essential for LOB, BER, and DEN detection (Fig. 1e). We next found misexpression of GR32a, GR59c, and GR66a together confers sensitivity to LOB, BER, and DEN (Fig. 1f), but not CAF, STR, SOA, or UMB. This suggests GR32a/GR59c/GR66a together form

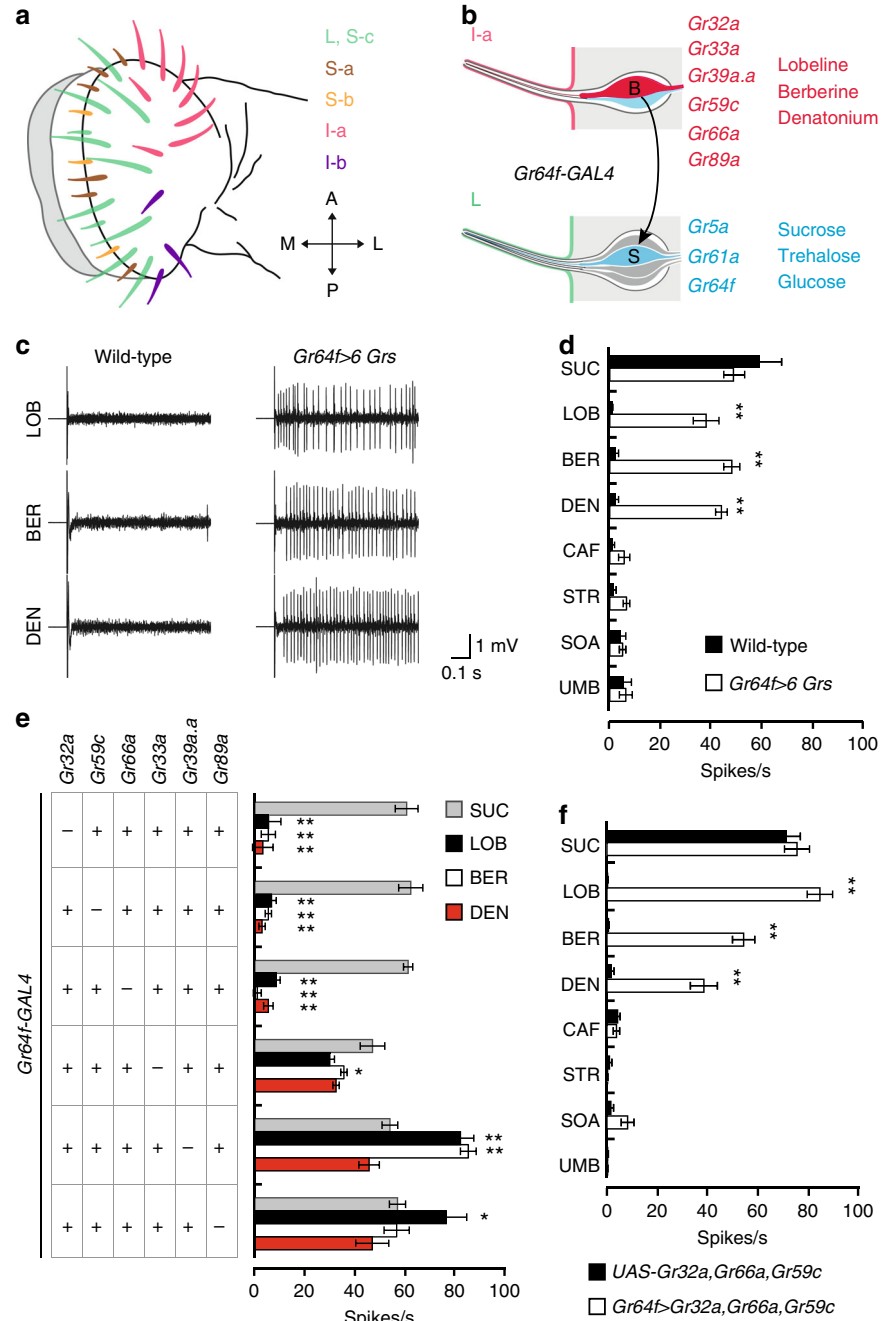

**Fig. 1** Identification of *Grs* for LOB, BER, and DEN detection. **a** Spatial localization of each class of taste bristle in the labellum. Figure modified from Weiss et al.[3] (2011). **b** Schematic for the misexpression of bitter *Grs* from I-a sensilla in the sweet GRNs of L-type sensilla using *Gr64f-GAL4*. The *Grs* normally expressed in bitter GRNs of I-a sensilla and their bitter ligands (red), as well as the sweet *Grs* normally expressed in sweet GRNs of L-type sensilla and their sugar ligands (blue) are indicated. **c** Representative traces from L-type sensilla of flies of the indicated genotypes upon application of 1 mM stimuli of LOB, BER, or DEN. **d** Mean responses of L-type sensilla induced by the indicated chemicals: 100 mM sucrose (SUC), 1 mM LOB, 1 mM BER, 1 mM DEN, 5 mM CAF, 1 mM STR, 1 mM SOA, and 1 mM UMB. $n = 6-10$. Unpaired *t*-test or the Mann−Whitney *U*-tests as appropriate. **e** Identification of the *Grs* required for LOB, BER, and DEN detection. The mean responses of L-type sensilla of the indicated genotypes upon application of 1 mM stimuli of LOB, BER, and DEN as well as 100 mM SUC are shown. $n = 6-10$. ANOVAs followed by Dunnet T3 post hoc tests for LOB and the Kruskal−Wallis tests followed by Mann −Whitney *U* post hoc tests for BER and DEN. The asterisks indicate statistically significant differences from *Gr64f >6 Grs* flies in **d**. **f** The mean responses of L-type sensilla expressing *Gr32a*, *Gr59c*, and *Gr66a* induced by the indicated chemicals. $n = 6-10$. Unpaired Student's *t*-tests or the Mann−Whitney *U*-tests as appropriate. All data are presented as means ± S.E.M. *$p < 0.05$, **$p < 0.01$. Complete genotypes are listed in Supplementary Table 6

a receptor for LOB, BER, and DEN. Since neither the expression of any one of these three *Grs* nor any combination of two of these three *Grs* (i.e., *Gr32a/Gr66a*, *Gr32a/Gr59c*, or *Gr66a/Gr59c*) is sufficient to confer LOB, BER, and DEN sensitivity, GR32a, GR59c, and GR66a are the minimal and essential components for

LOB, BER, and DEN detection (Supplementary Fig. 1b). It is noteworthy that misexpression of *Gr32a*, *Gr59c*, and *Gr66a* did not affect the sucrose responses of sweet GRNs, even as it conferred LOB, BER, and DEN responses (Fig. 1d–f). Furthermore, misexpression of *Gr32a*, *Gr59c*, and *Gr66a* in the sweet

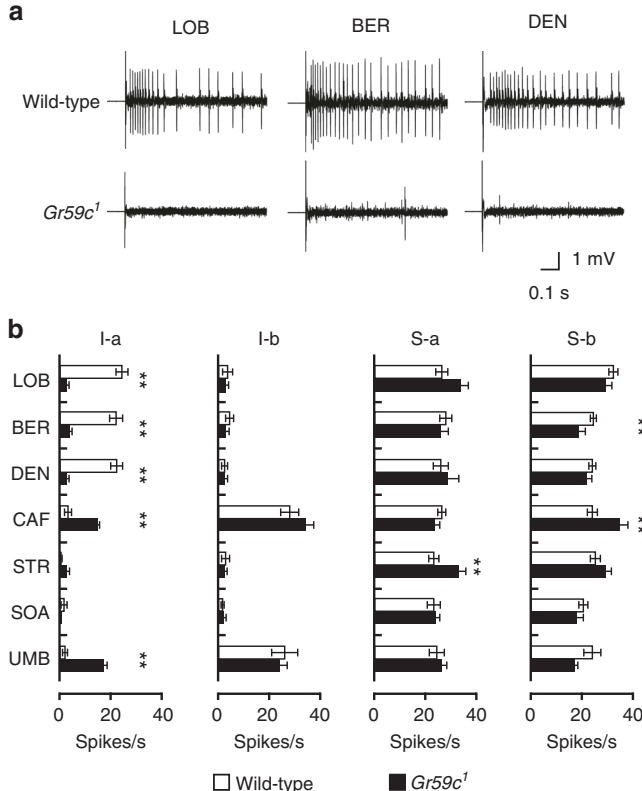

**Fig. 2** Electrophysiological responses of $Gr59c^1$ mutant sensilla to bitter chemicals. **a** Representative traces from I5 sensilla from wild-type and $Gr59c^1$ flies upon application of 1 mM stimulations with the indicated chemicals. **b** Mean responses of the indicated sensilla in wild-type and $Gr59c^1$ flies to 1 mM stimuli of the indicated bitter compounds except 5 mM CAF. $n = 6$–50. Unpaired Student $t$-tests or Mann–Whitney $U$-tests as appropriate. All data are presented as means ± S.E.M. **$p < 0.01$

GRNs of a new $Gr64$ cluster deletion mutant ($Gr64af$) covering the coding regions of the entire $Gr64$ cluster conferred a similar level of sensitivity to LOB, BER, and DEN as that in wild-type sweet GRNs (Supplementary Figs. 1c, 2a). Previous mutants of the $Gr64$ cluster either deleted additional neighboring genes or only deleted some genes of the cluster[22–25], and were thus less suitable for this purpose. Loss of six of the nine sweet clade $Grs$ does not affect ectopic responses to LOB, BER, and DEN upon $Gr32a$, $Gr59c$, and $Gr66a$ expression in sweet GRNs. This indicates that the ectopic responses to LOB, BER, and DEN we observed are not due to erratic interactions between endogenous sweet clade GRs and misexpressed bitter GRs.

**I-a sensilla require $Gr59c$ for bitter compound response.** None of the labellar sensilla of $Gr32a$ and $Gr66a$ mutant flies show a significant electrophysiological response to LOB, BER, or DEN (Supplementary Fig. 1d)[15]. We asked whether loss of $Gr59c$ also impairs sensitivity to LOB, BER, and DEN by generating a $Gr59c$ mutant ($Gr59c^1$) by ends-out homologous recombination (Supplementary Fig. 2f)[26]. As expected, the I-a sensilla of $Gr59c$ mutant flies do not respond to LOB, BER, or DEN (Fig. 2a, b). The S-a and S-b sensilla of $Gr59c^1$ flies, which do not express $Gr59c$, show wild-type responses to LOB, BER, and DEN. This suggests other receptor complexes are responsible for LOB, BER, and DEN sensitivity in these sensilla (Fig. 2b). In addition, we found $Gr59c^1$ flies show enhanced responses to CAF, STR, and UMB (Fig. 2b). This is consistent with a previous report that the I-a sensilla of $Gr59c$ mutant flies show increased sensitivity to

CAF and UMB[21]. In addition to confirming this, we found loss of $Gr59c$ also increases the sensitivity of S-a sensilla to STR and S-b sensilla to CAF.

**Identifying $Grs$ for bitter compound sensing in S-b sensilla.** S-b sensilla detect LOB, BER, and DEN in the absence of $Gr59c$, suggesting another GR complex recognizes these compounds in these sensilla. We, therefore, set out to identify the bitter $Grs$ required for LOB, BER, and DEN detection in S-b sensilla. Bitter GRNs in S-b sensilla express 16 different $Grs$[3]. While $Gr32a$, $Gr33a$, and $Gr66a$ are required for the detection of LOB, BER, and DEN[15, 19], $Gr8a$ is not[14]. To investigate the rest of these 16 $Grs$, we obtained three $Gr$ mutants that cover five $Grs$ (i.e., $Gr22e$, $Gr28b.a$, $Gr28b.d$, $Gr28b.e$, $Gr36c$)[27] and then generated five more mutants (i.e., $Gr22f^1$, $Gr28a^1$, $Gr36a^1$, $Gr39b^1$, $Gr89a^1$) by ends-out homologous recombination (Supplementary Fig. 2)[26]. We then examined the responses of their S-b sensilla to LOB, BER, and DEN. We were forced to exclude $Gr39a.a$ and $Gr59a$; the $Gr39a.a$ mutant is adult-lethal, and our attempts at obtaining a $Gr59a$ mutant were unsuccessful.

Consistent with previous reports[15, 19], loss of $Gr32a$, $Gr33a$, or $Gr66a$ nearly abolishes LOB, BER, and DEN-evoked action potentials in S-b sensilla (Fig. 3a–c). In addition, we found the S-b sensilla of $Gr22e$ mutant flies show significantly reduced responses to LOB, BER, and DEN (Fig. 3a–d), whereas those of $Gr22f$ and $Gr39b$ mutants show selectively reduced responses to DEN (Fig. 3c).

**S-b sensilla required $Gr22e$ for bitter compound response.** Next, we sought a more detailed picture of the role $Gr22e$ plays in the detection of aversive compounds. In addition to their reduced sensitivity to LOB, BER, and DEN, $Gr22e$ mutant S-b sensilla show lower sensitivity to STR and SOA (Fig. 3e). $Gr22e$ mutant S-a sensilla are less sensitive to DEN and STR (Fig. 3e).

Since S-b sensilla lacking $Gr22e$ show reduced responses to LOB, BER, DEN, STR, and SOA, we asked whether GR22e, together with GR32a and GR66a, forms the functional receptor for these bitter compounds. Misexpression of $Gr22e$, $Gr32a$, and $Gr66a$ using $Gr64f$-GAL4 does not alter sucrose responses but it does confer sensitivity to LOB, BER, DEN, and STR on the sweet GRNs of L-type sensilla (Fig. 3f). We also found ectopic co-expression of $Gr22e$, $Gr32a$, and $Gr66a$ in sweet GRNs lacking the $Gr64$ cluster ($Gr64af$) confers sensitivity to LOB, BER, DEN, and STR, indicating GR22e/GR32a/GR66a represents the minimal complex necessary for LOB, BER, DEN, and STR responses (Supplementary Fig. 3a). Since loss of $Gr22f$ and $Gr39b$ reduces the responses of S-b sensilla to DEN, we also asked whether $Gr22f$ and $Gr39b$ participate in the DEN receptor. Similar misexpression of either $Gr22f$ or $Gr39b$ with both $Gr32a$ and $Gr66a$ does not confer DEN sensitivity on the sweet GRNs of L-type sensilla (Supplementary Fig. 3b). This suggests the reduced responses to DEN in the $Gr22f$ and $Gr39b$ mutants are non-specific. They may instead be due to the formation of a DEN-detecting complex with other GRs or to an indirect reduction in DEN responses.

**$Gr22e$ and $Gr59c$ are redundant in bitter compound detection.** Since $Gr59c$ and $Gr22e$ are expressed in different sensilla and independently participate in receptor complexes that each respond to LOB, BER, and DEN, we wondered whether $Gr22e$ can substitute for $Gr59c$ in I-a sensilla and vice versa in S-b sensilla. We, therefore, expressed wild-type $Gr22e$ or $Gr59c$ in bitter GRNs of $Gr22e$ mutants using $Gr33a$-GAL4. As expected, $Gr22e$ expression rescues the $Gr22e$ mutant phenotype in S-b sensilla (Fig. 4a–d). In addition, $Gr22e$ and $Gr59c$ expression both induce hypersensitivity to LOB, BER, and DEN in S-a and S-b

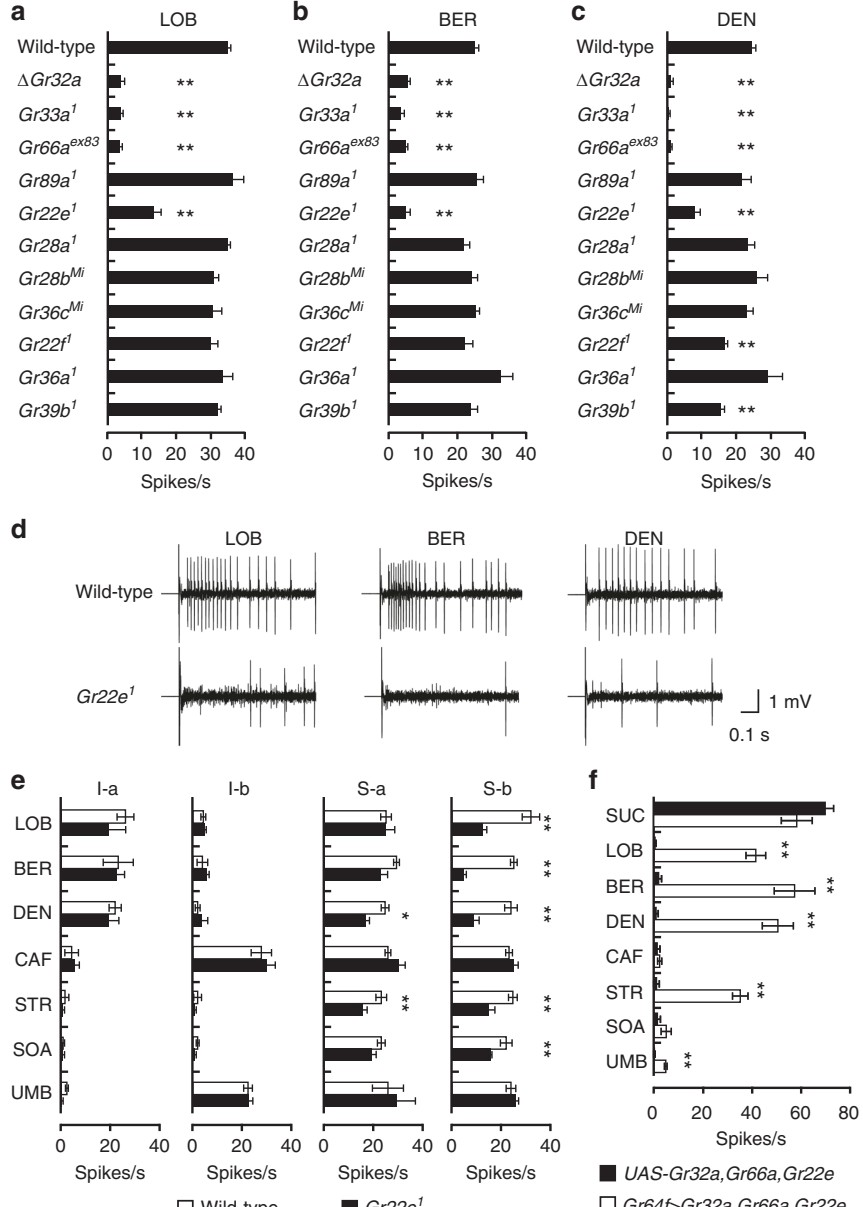

**Fig. 3** Identification of *Grs* for LOB, BER, and DEN detection in S-b sensilla. **a**–**c** Screening to identify the *Grs* required for the responses in S-b sensilla to LOB **a**, BER **b**, and DEN **c**. 1 mM of each chemical was used. *n* = 8–31. ANOVAs with Tukey or Dunnett T3 post hoc tests or Kruskal–Wallis tests with Mann–Whitney *U* post hoc tests as appropriate. The asterisks indicate statistically significant differences from wild-type flies. **d** Representative traces from wild-type and *Gr22e¹* S5 sensilla upon application of LOB, BER, and DEN. **e** Mean responses of the indicated sensilla from wild-type and *Gr22e¹* flies to 1 mM stimulations with the indicated bitter substances except 5 mM CAF. *n* = 6–30. Unpaired Student's *t*-tests or Mann–Whitney *U*-tests as appropriate. **f** Mean responses of Gr32a, Gr66a, and Gr22e-expressing L-type sensilla induced by 1 mM stimulations with the indicated chemicals except 100 mM SUC and 5 mM CAF. *n* = 7–14. Unpaired Student's *t*-tests or Mann–Whitney *U*-tests as appropriate. All data are presented as means ± S.E.M. *\*p* < 0.05, *\*\*p* < 0.01. Complete genotypes are listed in Supplementary Table 6

sensilla (Fig. 4a–c). Only *Gr22e*, however, rescues the STR detection defect of *Gr22e¹* S-b sensilla and induces STR hypersensitivity in S-a sensilla (Fig. 4d). This suggests the GR22e/GR32a/GR66a complex detects STR, but the GR59c/GR32a/GR66a complex does not. Neither misexpression of *Gr22e* nor overexpression of *Gr59c* affects sensitivity to LOB, BER, and DEN in I-a sensilla (Fig. 4a–c), but misexpression of *Gr22e* results in a novel response to STR in I-a sensilla (Fig. 4d). Misexpression of *Gr22e* or *Gr59c* in I-b sensilla results in a novel response to LOB, BER, and DEN and misexpression of *Gr22e* also results in a novel response to STR in I-b sensilla (Fig. 4a–d).

Next, we expressed either wild-type *Gr22e* or *Gr59c* in bitter GRNs of *Gr59c* mutants using *Gr33a-GAL4*. Misexpression of *Gr22e* or *Gr59c* in the S-a and S-b sensilla of *Gr59c* mutant flies leads to their hypersensitivity to LOB, BER, and DEN (Fig. 4e–g). Expression of either *Gr22e* or *Gr59c* can rescue the defect in LOB, BER, and DEN sensitivity observed in the I-a sensilla of *Gr59c* mutants (Fig. 4e–g). Misexpression of *Gr22e* or *Gr59c* in I-b sensilla results in a novel response to LOB, BER, and DEN (Fig. 4e–g). Only *Gr22e* overexpression induces STR hypersensitivity in S-a and S-b sensilla and only *Gr22e* misexpression induces novel responses to STR in I-a and I-b sensilla (Fig. 4h).

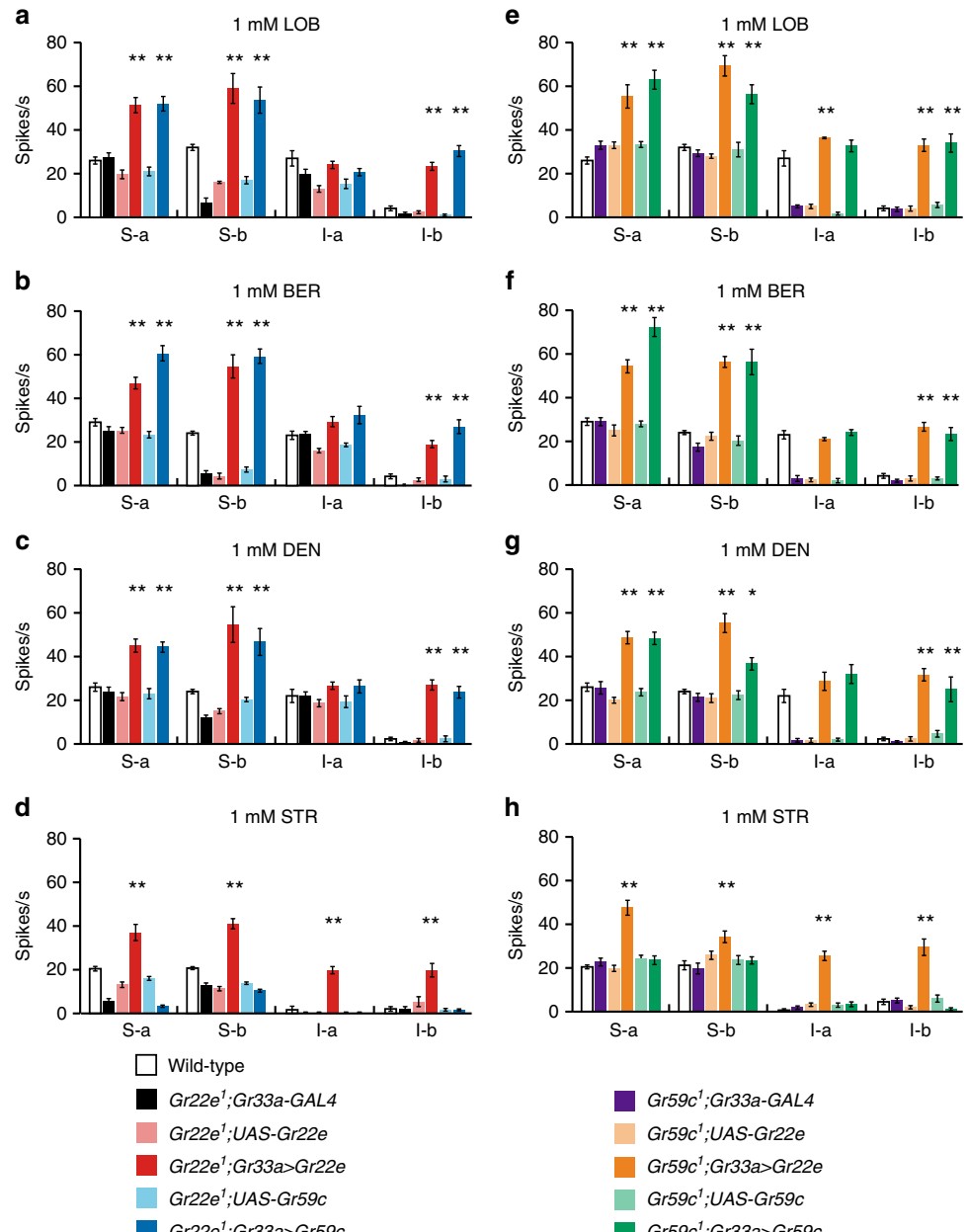

**Fig. 4** Functional redundancy of *Gr22e* and *Gr59c*. Mean responses in flies of the indicated genotypes evoked by 1 mM stimulations with LOB **a**, **e**, BER **b**, **f**, DEN **c**, **g**, or STR **d**, **h**. $n = 6$–69. ANOVAs followed by Tukey or Dunnett T3 post hoc tests or Kruskal–Wallis tests followed by Mann–Whitney $U$ post hoc tests as appropriate. All data are presented as means ± S.E.M. The asterisks indicate statistically significant differences (*$p < 0.05$, **$p < 0.01$) from wild-type flies. Complete genotypes are listed in Supplementary Table 6

In summary, expression of *Gr22e* in the *Gr22e* mutant and expression of *Gr59c* in the *Gr59c* mutant both rescue their respective mutant phenotypes, confirming the respective defects are attributable to loss of *Gr22e* or *Gr59c*. In addition, expression of *Gr22e* in the *Gr59c* mutant and *Gr59c* in the *Gr22e* mutant produce nearly identical responses to LOB, BER, and DEN. This indicates *Gr22e* can substitute for *Gr59c* and vice versa, except in the detection of STR.

**Conferring bitter chemical-induced currents on S2 cells**. To determine whether *Gr22e* and *Gr59c* are ion conducting subunits that confer bitter compound sensitivity, we co-expressed *Gr22e*, *Gr32a*, and *Gr66a* or *Gr32a*, *Gr59c*, and *Gr66a* in *Drosophila* S2 cells and performed whole-cell voltage-clamp recordings. Consistent with our in vivo gain-of-function results, we found both

*Gr* combinations confer sensitivity to LOB, BER, and DEN on S2 cells (Supplementary Fig. 4). These novel responses, which appear as outwardly rectifying currents in Supplementary Fig. 4, also seem to be dose-dependent. GR22e/GR32a/GR66a-expressing S2 cells also respond to STR, but neither GR22e/GR32a/GR66a-expressing S2 cells nor GR59c/GR32a/GR66a-expressing S2 cells respond to CAF or SOA (Supplementary Fig. 4e, j). These results are consistent with our in vivo loss-of-function and misexpression results.

**Bitter compound detection in *Gr22e Gr59c* double mutants**. The S-a sensilla of both *Gr22e* and *Gr59c* single mutants show normal responses to LOB, BER, and DEN (Figs. 2, 3). We initially assumed this was a result of the functional redundancy of *Gr22e* and *Gr59c* in the detection of LOB, BER, and DEN in the S-a

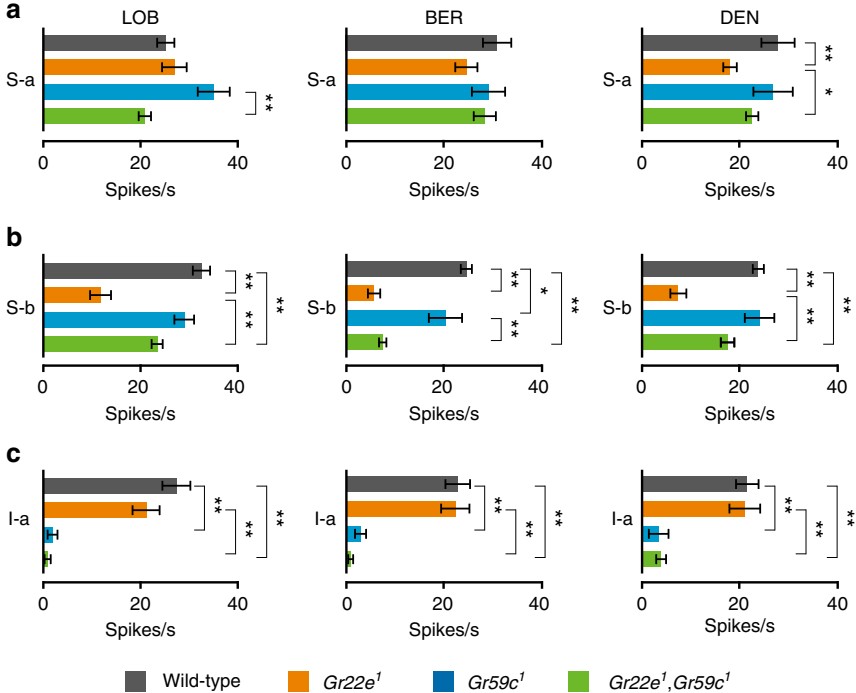

**Fig. 5** Effect of *Gr22e* and *Gr59c* mutation on S-a sensilla. Mean response of S-a sensilla of the indicated genotype to 1 mM stimulations with LOB **a**, BER **b**, and DEN **c**. n = 6–54. ANOVAs followed by Tukey or Dunnett T3 post hoc tests or Kruskal–Wallis tests followed by Mann–Whitney U post hoc tests as appropriate. All data are presented as means ± S.E.M. The asterisks indicate statistically significant differences (*p < 0.05, **p < 0.01)

sensilla. To test this possibility, we generated the *Gr22e,Gr59c* double mutant. When we measured the responses of the S-a sensilla to LOB, BER, and DEN, we found the S-a sensilla in the *Gr22e,Gr59c* double mutant does not show a significant reduction in response to LOB, BER, and DEN compared to the wild-type or single mutant with the exception of the comparison between the *Gr22e,Gr59c* double mutant and the *Gr59c* mutant in response to LOB (Fig. 5a). We were particularly surprised to note that while *Gr22e* mutant S-a sensilla show reduced responses to DEN, those of the *Gr22e,Gr59c* double mutant do not (Fig. 5a). These results suggest other *Grs* must play a role in the detection of LOB, BER, and DEN in S-a sensilla.

Next, we measured the LOB, BER, and DEN sensitivity of the S-b and I-a sensilla of the *Gr22e,Gr59c* double mutants. We found *Gr22e,Gr59c* double mutant S-b sensilla show reduced sensitivity to LOB, BER, and DEN like *Gr22e* single mutant S-b sensilla, but their defect in LOB and DEN detection is not as severe (Fig. 5b). *Gr22e,Gr59c* double mutant I-a sensilla, like those of the *Gr59c* single mutant, do not respond to LOB, BER, or DEN (Fig. 5c).

**Differential contribution of labellar sensilla to avoidance.** Finally, we used a two-way choice assay to ask how the different types of labellar bitter-responsive sensilla contribute to feeding decisions. We gave flies a choice between 1 mM sucrose and 5 mM sucrose combined with different concentrations of aversive compounds (LOB, BER, and DEN). While wild-type flies demonstrate dose-dependent repulsion to all the bitter compounds we tested (Fig. 6), both *Gr22e* and *Gr59c* single mutant flies show reduced repulsion to LOB- or DEN-containing foods (Fig. 6a, c). This defect is even more severe in the *Gr22e,Gr59c* double mutant (Fig. 6a, c). These data indicate the respective contributions of the S-b and I-a sensilla to behavioral LOB and DEN aversion are independent and additive. Surprisingly, we found that while the *Gr59c* mutant demonstrates a significant defect in BER avoidance, the *Gr22e* mutant does not (Fig. 6b).

The *Gr22e,Gr59c* double mutant, however, shows an even more severe defect in BER avoidance than the *Gr59c* single mutant (Fig. 6b). These data further suggest the gustatory system uses different types of sensilla expressing different receptors to direct subtly different behavioral responses depending on the bitter compound in question.

**Discussion**

Here we have clarified several principles that inform our understanding of the *Drosophila Grs* that detect aversive compounds. First, three *Grs* seem to cooperate to form functional bitter compound receptors. We have shown in sweet GRNs and *Drosophila* S2 cells that co-expression of *Gr22e*, *Gr32a*, and *Gr66a* is sufficient to confer sensitivity to LOB, BER, DEN, and STR, while co-expression of *Gr32a*, *Gr59c*, and *Gr66a* confers sensitivity to LOB, BER, and DEN. This is reminiscent of our identification of the heterotrimeric L-canavanine receptor—GR8a, GR66a, and GR98b[20]. If we arrange these bitter GR complex components in decreasing order of tuning breadth from broad to narrow, the resulting order would be: *Gr66a*, *Gr32a*, *Gr22e*, and *Gr59c*. Second, although *Drosophila* GR complexes function as heteromultimers, they seem to remain promiscuous. A given GR complex can be activated by multiple bitter chemicals: GR22e/GR32a/GR66a by LOB, BER, DEN, and STR; GR32a/GR59c/GR66a by LOB, BER, and DEN. Conversely, the same bitter chemicals can activate several different Gr complexes. For example, BER activates GR22e/GR32a/GR66a, GR32a/GR59c/GR66a, and another unknown GR complex in S-a sensilla. Finally, different GRNs use distinct GR combinations even for detecting the same chemicals. S-b sensilla require GR22e/GR32a/GR66a to detect LOB, BER, and DEN, while I-a sensilla require GR32a/GR59c/GR66a to accomplish the same thing. Since the S-a sensilla of *Gr22e,Gr59c* double mutants remain sensitive to LOB, BER, and DEN, they must have at least one additional receptor complex for these aversive compounds. It is likely S-b sensilla also

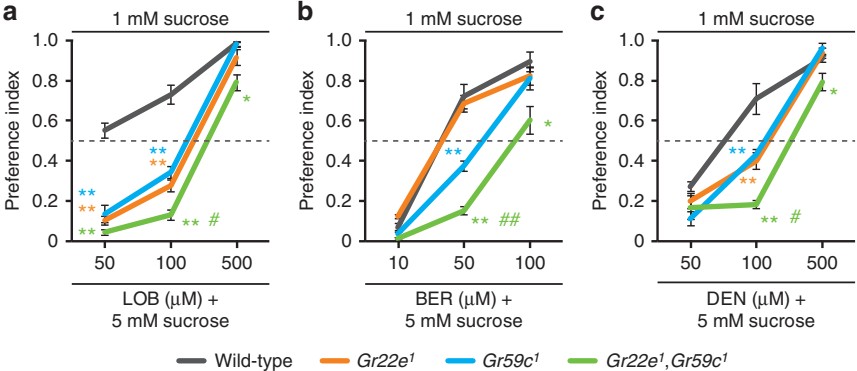

**Fig. 6** The contribution of *Gr22e* and *Gr59c* to bitter food aversion. Two-way choice assays using the indicated concentrations of **a** LOB, **b** BER, and **c** DEN. *n* = 4–12. ANOVAs followed by Tukey post hoc tests or Kruskal−Wallis tests followed by Mann−Whitney U post hoc tests as appropriate. All data are presented as means ± S.E.M. The asterisks and sharps indicate statistically significant differences from wild-type flies (*$p < 0.05$, **$p < 0.01$) and from both *Gr22e[1]* and *Gr59c[1]* (#$p < 0.05$, ##$p < 0.01$), respectively

have another receptor complex that responds to LOB, BER, and DEN, because S-b sensilla in the *Gr22e* mutant still respond weakly to these chemicals and the S-b sensilla of *Gr22e,Gr59c* double mutants are more sensitive to these chemicals than those of the *Gr22e* single mutant, presumably due to increase of expression of unknown LOB, BER, and DEN receptor complexes in the *Gr22e,Gr59c* double mutants (Fig. 5b).

The fact that *Gr22e,Gr59c* double mutant S-a sensilla show a near wild-type response to LOB, BER, and DEN (Fig. 5) makes it clear that distinct bitter GR complexes (even in the same GRN) can detect overlapping sets of bitter compounds. What advantage does such a strategy provide? We speculate that this permits specific sensilla to fine tune their responses to bitter compounds. For example, increasing the expression of *Gr22e* in S-a or S-b sensilla would make them more sensitive to LOB, BER, DEN, and STR. As expected with such a scheme, loss of either *Gr22e* or *Gr59c* results in a partial impairment of LOB, BER, and DEN avoidance rather than a complete loss of aversion (Fig. 6). Although we could not exclude the possibility that the behavioral consequences of the loss of *Gr22e* or *Gr59c* were influenced by taste organs other than the labellum (e.g., the tarsal taste sensilla) (Fig. 6), this is unlikely because the *Gr22e* and *Gr59c* reporters are not expressed in the tarsal taste sensilla[28]. This raises the interesting possibility that distinct GR complexes sense bitter chemicals not only depending on the taste sensilla in which they are expressed but also the different taste organs.

A growing body of evidence suggests interactions between *Grs* co-expressed in bitter GRNs are an important determiner of the GRN's response profile[3, 21]. We found overexpression or misexpression of *Gr22e* or *Gr59c* in S-a and S-b sensilla increases the endogenous response to LOB, BER, and DEN (Fig. 4). It is likely the extra GR22e or GR59c proteins bind GR32a and GR66a. This would form more LOB, BER, and DEN receptor complexes, increasing the sensitivity of the response to these chemicals. In contrast, neither misexpression of *Gr22e* nor overexpression of *Gr59c* affects LOB, BER, and DEN responses in I-a sensilla. Misexpression of *Gr22e* or overexpression of *Gr59c* in the *Gr59c* mutant results in a complete wild-type level rescue. It is possible that in wild-type I-a sensilla, most GR32a and GR66a molecules are already bound to GR59c, and therefore extra GR22e or GR59c do not form additional LOB, BER, and DEN receptors with GR32a and GR66a. Since misexpression of *Gr22e* or *Gr59c* in I-b sensilla confers on them novel sensitivity to LOB, BER, and DEN, it is possible the exogenous GR22e or GR59c proteins are outcompeting other GRs for binding to the endogenous GR32a and GR66a molecules, forming novel LOB, BER, and DEN receptor

complexes. If this hypothesis is true, we should see a concurrent reduction in the responses of these I-b sensilla to their normal ligands. Indeed, overexpression of *Gr59c* suppresses the normal I-b response to CAF, presumably secondary to a reduction in the number of functional CAF receptor complexes[3, 21]. In addition, loss of *Gr59c* in I-a sensilla, which shifts their bitter response profile to one resembling that of I-b sensilla (Fig. 2b)[21], allows other GRs expressed in I-a sensilla to form new functional receptor complexes. This then alters their sensitivity to CAF, UMB, TPH, and other bitter compounds (Fig. 2b)[21].

In summary, our results suggest bitter coding is much more complex and dynamic than expected. In the future, we hope to correlate changes in *Gr* expression with changes in environmental conditions and internal physiological states to better understand the dynamics of this system and how they affect animal behavior.

## Methods

**Fly stocks**. All fly stocks were maintained on standard cornmeal-molasses agar medium at 25 °C and 60% humidity under a 12 h/12 h light/dark cycle, respectively. 70FLP,70I-SceI/CyO (BL6934), Gr28b[mi] (BL24190), and Gr36c[mi] (BL26596) were obtained from the Bloomington *Drosophila* Stock Center. Gr22e[1] (#140936) was obtained from the Kyoto Stock Center. To minimize genetic background artifacts, all mutant strains were outcrossed to w[1118] for five generations. ΔGr32a was a gift from Hubert Amrein. Gr64f-GAL4 and UAS-Gr59c were gifts from John Carlson. Gr33a[1], Gr66a[ex83], UAS-Gr33a, UAS-Gr66a, and Gr33a-GAL4 were previously described[18, 19]. Gr64af harboring the deletion of the whole Gr64 cluster will be described in a separate manuscript in detail (Supplementary Fig. 2a).

**Generation of transgenic flies**. UAS-Gr22e, UAS-Gr22f, UAS-Gr32a, UAS-Gr39a. a, UAS-Gr39b, and UAS-Gr89a were generated. Labellar cDNAs were synthesized using Thermo Scientific RevertAid Reverse Transcriptase (Thermo Fisher Scientific, Waltham, MA) followed by transcript extraction using TRIZOL (Thermo Fisher Scientific). Individual *Gr* cDNAs were amplified and cloned from total labellar cDNAs using primer pairs listed in Supplementary Table 2. The amplified DNA fragments were inserted into pUAST vectors via conventional molecular cloning methods. pUAST vectors carrying each *Gr* cDNA were injected into w[1118] embryos by standard transgenesis techniques (BestGene Inc., Chino Hills, CA).

**Generation of *Gr* mutants**. Gr22f[1], Gr28a[1], Gr36a[1], Gr39b[1], Gr59c[1], and Gr89a[1] were generated by homologous recombination[26]. In general, 3 kb 5′ and 3′ homology arms for each *Gr* coding region were amplified by genomic DNA PCR using primer pairs listed in Supplementary Table 3 and cloned into mutant construction vectors. pw35 was used for Gr22f[1], Gr36a[1], Gr39b[1], and Gr89a[1]. pw35GAL4 and pw35+ were used for Gr28a[1] and Gr59c[1], respectively. After obtaining transformants carrying each targeting vector, the transgenes were mobilized by crossing to 70FLP,70I-SceI/CyO. Mosaic-eyed progeny (F1) were crossed to w[1118] to obtain red-eyed F2 progenies, which were subjected to PCR analysis[29]. Each *Gr* mutation was confirmed by genomic PCR using primers listed in Supplemental Table 4.

**Quantitative PCR**. Total RNA was extracted from dissected labella using TRIZOL (Thermo Fisher Scientific) according to the manufacturer's instructions. Labellar cDNA was synthesized from 1 μg of total RNA with the RevertAid reverse transcriptase system (Thermo Fisher Scientific). Quantitative PCR was conducted with a Quantstudio3 real-time PCR instrument (Thermo Fisher Scientific) using the ABI SYBR green system (Applied Biosystems, Foster City, CA). $C_T$ values were measured in triplicate. The sequence of the primer pairs for each *Gr* are shown in Supplementary Table 5. The concentrations of the linearized DNA carrying each of 6 *Grs* were measured with a Nanodrop (Thermo Fisher Scientific) and a series of 10-fold dilutions of linearized DNA were subjected to quantitative PCR analysis to establish standard curves and determine their regression equation. The average $C_T$ value for each target gene obtained from the same genotype were then used to calculate the absolute copy numbers for each *Gr* transcript in labella of the indicated genotypes. Absolute copy number of the target gene = (copy number of the standard) $\times 10^{(Ct-b)/m}$, where b and m refer to the slope and intercept of the standard curve regression equation, respectively. The number of transcripts in sweet GRNs was calculated by subtracting the number of transcripts in the *Gr64f-GAL4* control (endogenous bitter *Gr* expression level) from the number of transcripts of flies ectopically expressing *Grs*. RNA extraction and quantitative PCR were performed at least three times.

**Tip recording**. 2–5-day-old flies housed in fresh vials were anesthetized by brief ice exposure. A glass capillary pipette containing Ringer's solution was inserted through the thorax to the base of the labellum and connected to ground to serve as the reference electrode. The indicated concentrations of each tastant compound were dissolved in 30 mM tricholine citrate for electrical conductivity in a 10–20 μm diameter recording electrode. After the recording electrode was connected to a TastePROBE (Syntech, Hilversum, The Netherlands), the electrical signals derived from taste sensilla were recorded using an acquisition controller (Syntech) attached to a computer. These signals were amplified (×10), band-pass-filtered (100–3000 Hz), and sampled at 12 kHZ. Neuronal firing rates were then analyzed using the Autospike 3.1 software (Syntech).

**Two-way choice behavioral assay**. Briefly, 40–50 3–5-day-old flies were collected under $CO_2$ anesthesia at least 1 day prior to starvation. The flies were starved for 18 h in vials containing only 1% agarose gel. Then, they were given 90 min in a dark room at room temperature to choose between 1% agarose gel containing 1 mM sucrose or 5 mM sucrose and the indicated concentrations of bitter chemicals. To monitor their food preference, each food was mixed with either a blue (0.125 mg/ml Brilliant blue FCF) or red (0.2 mg/ml sulforhodamine B) non-toxic dye. After freezing the flies, the color of their abdomens was scored under a stereomicroscope. A preference index was calculated using the following equation: Preference Index (PI) = (# of red or blue abdomens + ½ the # of purple abdomens)/ Total # of fed flies.

**Whole-cell patch clamp recordings**. *Gr*- and EGFP-expressing S2 cells on coverslips were transferred to a chamber positioned on the stage of an inverted microscope (IX73, Olympus, Tokyo, Japan). Whole-cell currents were measured using a multiclamp amplifier at a holding potential of –60 mV (Axon Instruments, Foster City, CA) at room temperature. The bath solution contained normal Ringer's solution: 140 mM NaCl, 5 mM KCl, 2 mM $CaCl_2$, 2 mM $MgCl_2$, 10 mM HEPES (titrated to pH 7.4 with NaOH). The pipette solution contained 140 mM CsCl, 5 mM EGTA, 10 mM HEPES, 2 mM MgATP, 0.2 mM NaGTP (titrated to pH 7.2 with CsOH). Electrodes were pulled from borosilicate glass to a final resistance of 2–4 MΩ after fire-polishing. The seal resistances were 3–10 GΩ. After establishing a whole-cell configuration, currents were recorded in the presence of LOB, BER, DEN, CAF, STR, or SOA by applying 400 ms voltage-ramp pulses from –80 to +80 mV every 1500 ms. Currents were digitized with a Digidata 1440 A converter (Axon Instruments), filtered at 5 kHz, and analyzed using Clampfit 10.2 (Axon Instruments).

**Chemicals**. Berberine, caffeine, denatonium, lobeline, sucrose, sucrose octaacetate, strychnine, sulforhodamine B, tricholine citrate, and umbelliferone were purchased from Sigma-Aldrich (Saint Louis, MO). Brilliant blue FCF was purchased from Wako Pure Chemical Industries, Ltd (Osaka, Japan).

**Statistics**. All data were subjected to the Levene and Kolmogrov–Smirnov tests to evaluate variance homogeneity and normality, respectively. Depending on the results of these tests, ANOVAs with Tukey or Dunnet T3 post hoc tests or Kruskal −Wallis tests with Mann−Whitney $U$ post hoc tests were used to identify significant differences in comparisons between multiple groups. The unpaired Student's *t*-test or the Mann−Whitney $U$-test was used for comparisons between two groups.

**Data availability**. All relevant data are available from the authors upon request.

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

## Acknowledgements

We thank the Bloomington Stock Center and Drs H. Amrein and J. Carlson for fly stocks. We thank Dr J. Shim, Mr. J. Choi, Ms. H. S. Choi, and Miss H. Na for helping generate mutants and transgenic flies. This work was supported by a National Research Foundation of Korea (NRF) Grant funded by the Korean Government (MSIP) (NRF-2016R1A5A2008630 and NRF-2012M3A9B2052524 to S.J.M., NRF-2016R1D1A1B03932743 to J.Y.K.).

## Author contributions

H.Y.S. performed the in vivo electrophysiological experiments and analyzed the resulting data. Y.T.J. performed the behavioral assays and analyzed the resulting data. H.K. and S.M.O helped with the experiments for the revised version of this manuscript. J.Y.L. and S.W.H. performed the patch clamp experiments and analyzed the resulting data. J.Y.K. and S.J.M. supervised the project and wrote the paper.

## Additional information

**Competing interests:** The authors declare no competing financial interests.

