## [Peer Review File · Nature Communications]

Reviewers' comments:

Reviewer #1 (Remarks to the Author):

In the manuscript "Heterogeneity in the *Drosophila* gustatory receptor complexes that detect aversive compounds", Sung et al described an impressive tour de force on how combinations of bitter GRs confer specific sensitivity to bitter tastants. The authors conducted systematic gain- and loss-of function assays to interrogate the sufficiency and necessity of individual bitter GRs. They provide convincing experimental data to illustrate that not all "core" bitter GRs (Gr32a, Gr33a, Gr39a.a, Gr66a and Gr89a) are necessary to confer sensitivity for certain bitter tastants. In addition, the authors provide compelling evidence to support that a combination of three GRs (GRs 32a/59c/66a or GRs 32a/66a/22e) is sufficient to confer sensitivity to bitter compounds LOB, BER and DEN. Complementing a recent beautiful study by the Carlson lab (eLife, 2016), findings from the current manuscript significantly advance our understanding of the molecular coding logic of bitter taste.

With that said, my comments on the manuscript are minor and mainly editorial. To improve clarity, the current manuscript will benefit significantly from professional editing. For example, in the abstract, it was not immediate clear that GRs 32a/66a/59c and GRs 32a/66a/59c both confer sensitivity to LOB, BER and DEN but the latter GR combination is also responsive to an additional bitter tastant, STR.

Specific comments:

Line 71: should reference Weiss et al, Neuron, 2011, by the end of the sentence.

Line 121: In the experiments shown in Figure 1, have the authors verified that all six GRs are expressed in the Gr64f GRNs? If not, the authors need to discuss the possibilities that not all GRs are expressed or expressed at the same levels in the heterologous expression system.

Line 248: In the two-choice behavioral assay shown in Figure 6, the authors need to discuss whether bitter GRNs in the leg may complicate the interpretation of their results.

Lines 296-297: The authors stated that increasing the expression of Gr22e in I-a sensilla would make them more sensitive to BON, BER, and DEN. However, the results shown in Fig 4a-4c did not support this interpretation.

Lines 327: The authors speculated that "Gr33a may act as an "insulator" that prevents random aggregation of GRs...". It is unclear how the authors come up with this statement as there is no reference or rationale to support the interpretation.

Reviewer #2 (Remarks to the Author):

This is a detailed analysis of bitter Gustatory Receptors using a molecular genetic approach. Bitter chemical avoidance is mediated by gustatory neurons present in most labellar and tarsal taste sensilla. Previous analysis by the Carlson and the Moon labs have shown that a group of ~ 35 to 40 Gr genes are expressed in bitter GRNs, but it was found that some

GRNs express only a few (~6) while others express as many as 30 of these putative bitter Gr genes. Moon and colleagues conduct an extensive electrophysiological analysis using a small panel of bitter chemicals to elucidate the composition of multimeric bitter taste receptors. One elegant approach that was used is to express combinations of bitter GR proteins in sweet taste GRNs, which, in wild type flies, do not respond to bitter compounds, and then assess the response profile of such neurons upon stimulation with bitter compounds.

While this study is extensive, it adds little new information for how these receptors function. The results largely confirm previous findings by the Carlson and the Moon labs (including one recently already published in Nat Com) such as the notion of significant overlap between the functions of specific Gr genes. It also confirms previous findings that functional receptors are multimers, likely composed of three different subunits. However there is nothing conceptually novel and the paper is more appropriate for a more specialized journal. The authors also promote the idea that Gr66a plays a central role in bitter taste and compare it to Orco in the olfactory system, which is in direct contradiction of a previous study by Moon (see below). Lastly, the authors fail to discuss a much more relevant comparison of their findings with that of sweet taste Gr complexes. There are many similarities, which they unfortunately ignore.

I have a few minor points that should be addressed for publication in a more appropriate journal:

It is curious that in Figure 1e (in contrast to Figure 1d), the authors do not show the response to sucrose when Gr32a, Gr6a and Gr59c are expressed in sugar neurons. It is important to show that this combination of bitter Gr proteins does not alter sugar response. It has been observed that interference between bitter and sugar Gr proteins can occur, due to formation of naturally not occurring complexes when expression of such receptors is forced into one cell type. Ideally, the authors should have used a sugar blind mutant background for these experiments, which is certainly challenging, but not impossible. The potential caveat of having different receptor types in the same cells should at least be discussed.

In Figure 5, the authors compare the response of three different bitter GRNs (s-a, S-b and I-a) in Gr22a, Gr59c and Gr22a/59c double mutants. Curiously, they find that double mutants respond much better to lobelline and denatonium than Gr22a single mutants (single Gr59c mutants had no effect on response to these chemicals). There is no explanation given why that is the case.

In the discussion, the authors propagate the idea that Gr66a plays a similar role in bitter taste as Orco in the olfactory system (obligatory subunit). This cannot be true due to data published by the senior author (Current Biology, 2006). In this paper, it was shown that Gr66a mutants respond behaviorally to many bitter compounds normally. Moreover, electrophysiological responses to numerous bitter compounds were not affected in an S-a type sensilla (S6). Thus, while many bitter GRNs are more sensitive to the loss of this receptor, some do not seem to be affected at all or not as much. Thus, the comparison to Orco is inappropriate. Rather, the authors should discuss their findings with those observed in

sweet taste, which have been extensively studied recently. There seem to be significant similar findings (overlap/redundancy in function of most Grs, but some being more critical for responses to many sugars; see Fujii et al, 2015; and Yavuz et al., 2016).

This in addition makes the finding less novel than what the authors pretend them to be.

Reviewer #3 (Remarks to the Author):

In this manuscript, Sung et al show a comprehensive set of observations demonstrating that cells which ectopically express a combination of 3 gustatory receptors can detect specific aversive compounds, whether these cells are gustatory cells normally responding to sugars or s2 cells. The triad of receptors proposed here is either Gr59c/Gr32a/Gr66a or Gr22e/Gr32a/Gr66a. This proposition is supported by:

(1) the ectopic expression of 6 Grs into cells expressing Gr64f showing that cells normally responding to sugar can respond to some bitter tastants (the 6 Grs are Gr32a, Gr33a, Gr39a.a, Gr59c, Gr66a, Gr89a)

(2) the use of mutants of 11 Grs from which electrophysiological responses to 3 bitter tastants at one concentration (lobeline, berberine, denatonium) were recorded from Sb sensilla; this set of observations demonstrates that mutations of Gr32a, Gr33a, Gr66a, Gr22e induce a reduction of the responses to lobeline, berberine and denatonium

(3) the comparison of how mutations of Gr59c, Gr22a (or both) impacts electrophysiological responses of 4 types of sensilla to lobeline, berberine, denatonium, (and strychnine); they also look at how either of these genes can restore the function affected by the mutations of each these genes; in addition, they test the impact of these mutations on the feeding behavior

(4) the expression of 3 Grs (Gr32a, Gr66a, Gr22e or Gr59c) into S2 cells and the recording of current-voltage relations in the presence or absence of 3 tastants.

This work is modelled after a previous paper "The full repertoire of Drosophila gustatory receptors for detecting an aversive compounds" Nat Comm 2015, where Moon and Jeong demonstrated that co-expressing Gr66a +Gr8a +Gr98b into cells responding to sugar conferred them the capacity to detect L-canavanine, and that whole cell currents could also be recorded in response to L-canavanine in S2 cells expressing these receptors.

As such, the current work is quite interesting but it is difficult to read and a bit confusing.

The conclusions that can be drawn from this study are limited by:

- The use of the Gal4 system which does not allow them to control the level of expression of each of the Grs placed within a UAS-Gr construction. It is implicitly assumed that each Gr is expressed evenly but is it really the case?? Would it be possible to check this directly?

- The choice of aversive stimuli: mainly 3 molecules are tested, 7 in total are evaluated. The authors should make clear how the choice of these tastants and of the 3 main ones has been made. Why strychnine has been tested in one case but not in the other? (cf figure 4d)

- As the number of aversive stimuli is reduced, it is not possible to infer from this study if the 3-4 Grs studied here represent "core" receptors that always need to be present or if other Grs are needed when it comes to detecting other molecules. This should be more clearly discussed.

- The authors only refer to studies done on the proboscis. They should make this clear because it seems that the requirement for Grs is not the same on the legs (is Gr66a really

needed on the legs?) or in the gustatory receptors of the foregut.

- Unless I am mistaken, they do not discuss the results shown in figure 2b, where the mutation of Gr59c induces the acquisition of a new taste (sensilla I-a: CAF, UMB; sensilla S-a: STR; sensilla S-b: CAF) while it was expected that a mutation of this gene should induce a loss of a response. Does this mean that other core receptors are needed for CAF, UMB and STR?? The only mention I found of this result is indirect when, at the end of the discussion they comment ref (20) (as they mention TPH which is not tested in this work). It is interesting that the authors find such an increase of activity since the work of Delventhal et al (ref 20) was plainly suggesting that in some cases, the loss of Gr induces a loss of activity and in other cases, not, and that the modification expected could not be predicted.

- I would appreciate also the authors to comment about the role (or not) of Gr47a. In a previous study, Moon & Montell showed that strychnine, one of the chemicals tested in this work, was detected thanks to Gr47a or at least that Gr47a was necessary. In figure 4d of this work, the data presented suggest that this is not the case: S-a and S-b sensilla can detect strychnine when Gr22e is expressed within the construction Gr22e1; Gr33a>Gr22e. But STR is poorly detected in wild type flies. How can this be explained?

- The authors should discuss plainly that the 3 Grs expressed in cells expressing Gr64f are not expressed in an "empty" cell, and that these Grs could "pair" also with the other Grs present in these cells (ie Gr64f, Gr5a, Gr61a).

In addition to these general remarks, here are a few suggestions:

- It would be interesting to provide a table of the constructions used. For example, the naming of the construction used in figure 1 is a bit obscure. I would be interested to know why 2 Gr64f Gal4 drivers are noted, on which chromosomes they are inserted and also if the authors have controlled the level of expression of these Grs by RT-PCR for example. Is the level of expression of 1 UAS-X driven by 1 Gal4 equivalent the level of expression of 5-6 UAS-X driven by 1 or 2 gal4?

- Figure 1f & figure 3 f: did you test the response to sucrose? Is it affected by your constructions? Did you test the other parent, ie Gr64f-Gal4?

- Figure 2 could be paired with figure 3e.

- The electrophysiological recordings (figure 3d, figure 2a LOB) show nice large spikes but also a fairly strong activity of another taste cell. What is this activity? Did you observe this repeatedly?

- Figure 4: Did you test STR with Gr59c1 constructions? Could you make this figure more explicit? In its present form, it looks quite confusing.

- Figure 5: the statistical tests were done by comparing the responses of the single or double mutants to that of the wild-type. However, it looks that for some situations, the responses of the double-mutants are higher than those of the single mutants. Are these differences statistically relevant? This is the case for : S-a: DEN, Sb: LOB, S-b: DEN. One would expect the double mutant to respond at least at the lowest level of the 2 single mutants.

- Suppl 4: the lower part is mislabeled Gr32a, Gr66a, Gr22e instead of Gr32a, Gr66a, Gr59c.

Reviewer #1:

In the manuscript “Heterogeneity in the *Drosophila* gustatory receptor complexes that detect aversive compounds”, Sung et al described an impressive tour de force on how combinations of bitter GRs confer specific sensitivity to bitter tastants. The authors conducted systematic gain- and loss-of function assays to interrogate the sufficiency and necessity of individual bitter GRs. They provide convincing experimental data to illustrate that not all “core” bitter GRs (Gr32a, Gr33a, Gr39a.a, Gr66a and Gr89a) are necessary to confer sensitivity for certain bitter tastants. In addition, the authors provide compelling evidence to support that a combination of three GRs (GRs 32a/59c/66a or GRs 32a/66a/22e) is sufficient to confer sensitivity to bitter compounds LOB, BER and DEN. Complementing a recent beautiful study by the Carlson lab (eLife, 2016), findings from the current manuscript significantly advance our understanding of the molecular coding logic of bitter taste.

With that said, my comments on the manuscript are minor and mainly editorial. To improve clarity, the current manuscript will benefit significantly from professional editing. For example, in the abstract, it was not immediate clear that GRs 32a/66a/59c and GRs 32a/66a/59c both confer sensitivity to LOB, BER and DEN but the latter GR combination is also responsive to an additional bitter tastant, STR.

Specific comments:

Line 71: should reference Weiss et al, Neuron, 2011, by the end of the sentence.

We have added the suggested reference.

Line 121: In the experiments shown in Figure 1, have the authors verified that all six GRs are expressed in the Gr64f GRNs? If not, the authors need to discuss the possibilities that not all GRs are expressed or expressed at the same levels in the heterologous expression system.

We conducted a qPCR analysis to confirm that all six *Grs* are misexpressed in our experimental conditions. We compared three different groups: *Gr64f-GAL4* (control), *Gr64f-GAL4>6 UAS-Grs*, and *2XGr64f-GAL4>6 UAS-Grs*. We found all six *Grs* are expressed reasonable levels in sweet GRNs. *GAL4* transgene copy number did not proportionally increase the expression level of each *Gr*. Now, we have rewritten the main text in page 6 to say “... confirmed the misexpression of each the *Gr* in the sweet GRNs by quantitative PCR (Supplementary Table 1).

Line 248: In the two-choice behavioral assay shown in Figure 6, the authors need to discuss whether bitter GRNs in the leg may complicate the interpretation of their results.

We agree. We added the following sentence to the Discussion. “Although we could not exclude the possibility that the behavioral consequence of the loss of Gr22e or Gr59c were influenced by taste organs other than the labellum, (e.g., the tarsal taste sensilla) (Fig. 6),

this is unlikely because the Gr22e and Gr59c reporters are not expressed in the tarsal taste sensilla”

Lines 296-297: The authors stated that increasing the expression of Gr22e in I-a sensilla would make them more sensitive to BON, BER, and DEN. However, the results shown in Fig 4a-4c did not support this interpretation.

We made a mistake here. We appreciate the reviewer catching this embarrassing error before publication. We have now changed “I-a sensilla” to “S-a or S-b sensilla.”

Lines 327: The authors speculated that “Gr33a may act as an “insulator” that prevents random aggregation of GRs...” It is unclear how the authors come up with this statement as there is no reference or rationale to support the interpretation. Now we accepted this recommendation and deleted this part.

We agree and have removed the offending paragraph.

Reviewer #2:

This is a detailed analysis of bitter Gustatory Receptors using a molecular genetic approach. Bitter chemical avoidance is mediated by gustatory neurons present in most labellar and tarsal taste sensilla. Previous analysis by the Carlson and the Moon labs have shown that a group of ~ 35 to 40 Gr genes are expressed in bitter GRNs, but it was found that some GRNs express only a few (~6) while others express as many as 30 of these putative bitter Gr genes. Moon and colleagues conduct an extensive electrophysiological analysis using a small panel of bitter chemicals to elucidate the composition of multimeric bitter taste receptors. One elegant approach that was used is to express combinations of bitter GR proteins in sweet taste GRNs, which, in wild type flies, do not respond to bitter compounds, and then assess the response profile of such neurons upon stimulation with bitter compounds.

While this study is extensive, it adds little new information for how these receptors function. The results largely confirm previous findings by the Carlson and the Moon labs (including one recently already published in Nat Com) such as the notion of significant overlap between the functions of specific Gr genes. It also confirms previous findings that functional receptors are multimers, likely composed of three different subunits. However there is nothing conceptually novel and the paper is more appropriate for a more specialized journal. The authors also promote the idea that Gr66a plays a central role in bitter taste and compare it to Orco in the olfactory system, which is in direct contradiction of a previous study by Moon (see below). Lastly, the authors fail to discuss a much more relevant comparison of their findings with that of sweet taste Gr complexes. There are many similarities, which they unfortunately ignore.

I have a few minor points that should be addressed for publication in a more appropriate journal:

It is curious that in Figure 1e (in contrast to Figure 1d), the authors do not show the response to sucrose when Gr32a, Gr6a and Gr59c are expressed in sugar neurons. It is important to show that this combination of bitter Gr proteins does not alter sugar response.

We confirmed that misexpression of bitter *Grs* in the sweet GRNs does not alter sucrose responses. We added these results to Figure 1e, 1f, and 3f and explained them in the main text (Please see page 7 and page 9).

It has been observed that interference between bitter and sugar Gr proteins can occur, due to formation of naturally not occurring complexes when expression of such receptors is forced into one cell type. Ideally, the authors should have used a sugar blind mutant background for these experiments, which is certainly challenging, but not impossible. The potential caveat of having different receptor types in the same cells should at least be discussed.

To exclude the possibility that bitter responses in sweet GRNs misexpressing bitter *Grs* were due to abnormal interactions between endogenous sweet *Grs* and ectopically expressed bitter *Grs*, we expressed bitter *Grs* in *Gr64* cluster-deleted sweet GRNs and measured bitter chemical-evoked action potentials. As expected, *Gr64* cluster deletion (*Gr64af*) abolishes sucrose-evoked nerve firings. When we expressed *Gr32a*, *Gr66a*, and *Gr59c* together in the sweet GRNs of *Gr64af*, we observed responses to LOB, BER, and DEN. Also, when the *Gr32a*, *Gr66a*, and *Gr22e* combination was expressed in the sweet GRNs of *Gr64af*, the responses to STR as well as LOB, BER, DEN were observed. Unfortunately, we were unable to test completely sugar blind flies due to the difficulty of obtaining the desired genotype via genetic crosses. The sweet GRNs of *Gr64af* may express *Gr5a* and *Gr61a*. We could not exclude the possibility of interactions between GR5a/GR61a and ectopically expressed bitter GRs, but this is highly unlikely. These results are consistent with our hypothesis that GR32a/GR66a/GR59c and GR32a/GR66a/GR22e can form functional receptors for LOB/BER/DEN and LOB/BER/DEN/STR, respectively. We added these results to Supplementary Figures 1c and 3a and explained them in the text. Please see page 7 and page 9.

The *Gr64af* flies harboring the deletion of only the *Gr64* cluster coding region will be described in detail in an independent manuscript.

In Figure 5, the authors compare the response of three different bitter GRNs (s-a, S-b and l-a) in *Gr22a*, *Gr59c* and *Gr22a/59c* double mutants. Curiously, they find that double mutants respond much better to lobelline and denatonium than *Gr22a* single mutants (single *Gr59c* mutants had no effect on response to these chemicals). There is no explanation given why that is the case.

This was mentioned in the Discussion.

“It is likely S-b sensilla also have another receptor complex that responds to LOB, BER, and DEN, because S-b sensilla in the Gr22e mutant still respond weakly to these chemicals and the S-b sensilla of Gr22e,Gr59c double mutants are more sensitive to these chemicals than those of the Gr22e single mutant, presumably due to an increase in the expression of unknown LOB, BER, and DEN receptor complexes in the Gr22e,Gr59c double mutants.”

In the discussion, the authors propagate the idea that Gr66a plays a similar role in bitter taste as Orco in the olfactory system (obligatory subunit). This cannot be true due to data published by the senior author (Current Biology, 2006). In this paper, it was shown that Gr66a mutants respond behaviorally to many bitter compounds normally. Moreover, electrophysiological responses to numerous bitter compounds were not affected in an S-a type sensilla (S6). Thus, while many bitter GRNs are more sensitive to the loss of this receptor, some do not seem to be affected at all or not as much. Thus, the comparison to Orco is inappropriate. Rather, the authors should discuss their findings with those observed in sweet taste, which have been extensively studied recently. There seem to be significant similar findings (overlap/redundancy in function of most Grs, but some being more critical for responses to many sugars; see Fujii et al, 2015; and Yavuz et al., 2016). This in addition makes the finding less novel than what the authors pretend them to be.

Following the *Gr66a* paper referred to by the reviewer (Current Biology, 2006), many studies of *Gr66a* function have led to additional insight. *Gr66a* is now known to be necessary for the response to many bitter chemicals including lobeline, papaverine, and strychnine (Lee et al., Avoiding DEET through insect gustatory receptors. Neuron, 2010), as well as L-canavanine (Lee et al., Gustatory receptors required for avoiding the insecticide L-canavanine. J of Neuroscience, 2012), umbelliferone (Poudel et al., Gustatory receptors required for sensing umbelliferone in *Drosophila melanogaster*. Insect Biochem Mol Biol., 2015), and coumarin (Poudel and Lee, Gustatory Receptors Required for Avoiding the Toxic Compound Coumarin in *Drosophila melanogaster*. Mol Cells, 2016). These results together indicate *Gr66a* is involved in the responses to most bitter chemicals detected by gustatory receptors. Nevertheless, it may be an oversimplification to assume that *Gr66a* is required for all bitter responses because limited Grs and limited bitter chemicals have been tested. As the reviewer suggested, we removed the comparison with Orco.

The papers the reviewer mentioned all provide essential information on *Drosophila* sugar sensing using mutants for the sugar Grs as well as overexpression of sugar Grs in the sweet GRNs of sugar blind flies. In particular, various combinations of sweet GR complexes show redundant and unique responses. However, the minimal subunit composition of sweet GRs is still unclear. No one has shown the functional expression of sweet GRs in a heterologous system. With this limited information about sweet Grs, we felt a direct comparison between bitter Grs and sweet Grs did not seem appropriate.

Reviewer #3:

In this manuscript, Sung et al show a comprehensive set of observations demonstrating that cells which ectopically express a combination of 3 gustatory receptors can detect specific aversive compounds, whether these cells are gustatory cells normally responding to sugars or s2 cells. The triad of receptors proposed here is either Gr59c/Gr32a/Gr66a or Gr22e/Gr32a/Gr66a. This proposition is supported by:

(1) the ectopic expression of 6 Grs into cells expressing Gr64f showing that cells normally responding to sugar can respond to some bitter tastants (the 6 Grs are Gr32a, Gr33a, Gr39a.a, Gr59c, Gr66a, Gr89a)

(2) the use of mutants of 11 Grs from which electrophysiological responses to 3 bitter tastants at one concentration (lobeline, berberine, denatonium) were recorded from Sb sensilla; this set of observations demonstrates that mutations of Gr32a, Gr33a, Gr66a, Gr22e induce a reduction of the responses to lobeline, berberine and denatonium

(3) the comparison of how mutations of Gr59c, Gr22a (or both) impacts electrophysiological responses of 4 types of sensilla to lobeline, berberine, denatonium, (and strychnine); they also look at how either of these genes can restore the function affected by the mutations of each these genes; in addition, they test the impact of these mutations on the feeding behavior

(4) the expression of 3 Grs (Gr32a, Gr66a, Gr22e or Gr59c) into S2 cells and the recording of current-voltage relations in the presence or absence of 3 tastants.

This work is modelled after a previous paper "The full repertoire of Drosophila gustatory receptors for detecting an aversive compounds" Nat Comm 2015, where Moon and Jeong demonstrated that co-expressing Gr66a +Gr8a +Gr98b into cells responding to sugar conferred them the capacity to detect L-canavanine, and that whole cell currents could also be recorded in response to L-canavanine in S2 cells expressing these receptors.

As such, the current work is quite interesting but it is difficult to read and a bit confusing. The conclusions that can be drawn from this study are limited by:

- The use of the Gal4 system which does not allow them to control the level of expression of each of the Grs placed within a UAS-Gr construction. It is implicitly assumed that each Gr is expressed evenly but is it really the case?? Would it be possible to check this directly?

We conducted a qPCR analysis to verify that all six *Grs* are misexpressed in our experimental condition in three different genotypes: *Gr64f-GAL4* (control), *Gr64f-GAL4>6 UAS-Grs*, and *2XGr64f-GAL4>6 UAS-Grs*. We found all six *Grs* are expressed at reasonable levels in sweet GRNs. We found that *GAL4* transgene copy number does not proportionally increase the expression level of each *Gr*. Now, we have rewritten the main text on page 6 to indicate that we "... confirmed the misexpression of each the *Gr* in the sweet GRNs by quantitative PCR (Supplementary Table 1)."

- The choice of aversive stimuli: mainly 3 molecules are tested, 7 in total are evaluated. The authors should make clear how the choice of these tastants and of

the 3 main ones has been made. Why strychnine has been tested in one case but not in the other? (cf figure 4d)

We explained the reason we mainly tested three bitter chemicals at the beginning of the Results, page 6.

Originally, we did not test strychnine in *Gr59c* mutants because *Gr59c*¹ did not show any defect in strychnine responses. Nevertheless, for the sake of completeness, we conducted the suggested experiment and added the results to Figure 4h. We revised the text on page 11 as follows: "Only *Gr22e* overexpression induces STR hypersensitivity in S-a and S-b sensilla and only *Gr22e* misexpression induces novel responses to STR in I-a and I-b sensilla (Fig. 4h)."

- As the number of aversive stimuli is reduced, it is not possible to infer from this study if the 3-4 Grs studied here represent "core" receptors that always need to be present or if other Grs are needed when it comes to detecting other molecules. This should be more clearly discussed.

We do not suggest all the Grs studied here are core receptors. Especially, *Gr22e* and *Gr59c* appear to confer ligand specificity as shown in Figures 1f and 3f. In contrast, *Gr66a* is required for the detection of most bitter chemicals in combination with other bitter Grs. This is why we propose *Gr66a* may play a co-receptor role. Nevertheless, it may be an oversimplification to assume that *Gr66a* is required for all bitter responses because limited Grs and limited bitter chemicals have been tested. So we removed the comparison with Orco as well as co-receptor hypothesis.

- The authors only refer to studies done on the proboscis. They should make this clear because it seems that the requirement for Grs is not the same on the legs (is *Gr66a* really needed on the legs?) or in the gustatory receptors of the foregut.

We have now included the following in the Discussion. "Although we could not exclude the possibility that the behavioral consequences of the loss of *Gr22e* or *Gr59c* were influenced by taste organs other than the labellum (e.g., the tarsal taste sensilla) (Fig. 6), this is unlikely because the *Gr22e* and *Gr59c* reporters are not expressed in the tarsal taste sensilla. This raises the interesting possibility that distinct GR complexes sense bitter chemicals not only depending on the taste sensilla in which they are expressed but also the different taste organs."

- Unless I am mistaken, they do not discuss the results shown in figure 2b, where the mutation of *Gr59c* induces the acquisition of a new taste (sensilla I-a: CAF, UMB; sensilla S-a: STR; sensilla S-b: CAF) while it was expected that a mutation of this gene should induce a loss of a response. Does this mean that other core receptors are needed for CAF, UMB and STR?? The only mention I found of this result is indirect when, at the end of the discussion they comment ref (20) (as they mention TPH which is not tested in this work). It is interesting that the authors find such an increase of activity since the work of Delventhal et al (ref 20) was plainly suggesting

that in some cases, the loss of Gr induces a loss of activity and in other cases, not, and that the modification expected could not be predicted.

The gain-of-function experiment in the *Gr59c* mutant was discussed in the 'Interactions between *Drosophila* bitter GRs' section of the Discussion. Unknown GRs for CAF, UMB, and STR cannot form a complex with GR66a for the detection of CAF, UMB, and STR when GR59c is present; They only form functional receptor complexes with GR66a in the absence of GR59c. We have also added the reference for TPH.

- I would appreciate also the authors to comment about the role (or not) of Gr47a. In a previous study, Moon & Montell showed that strychnine, one of the chemicals tested in this work, was detected thanks to Gr47a or at least that Gr47a was necessary. In figure 4d of this work, the data presented suggest that this is not the case: S-a and S-b sensilla can detect strychnine when Gr22e is expressed within the construction Gr22e1; Gr33a>Gr22e. But STR is poorly detected in wild type flies. How can this be explained?

One of main findings of this manuscript is that distinct bitter GR complexes can detect the same bitter chemicals even in the same sensilla. A previous study showed *Gr47a* is required for strychnine sensing in S-a and S-b sensilla. Nevertheless, in the current study, we found *Gr22e* partially contributes to strychnine sensing in S-a and S-b sensilla and that the GR32a/GR66a/GR22e complex is sufficient for strychnine detection (Please see Fig. 3f and Supplementary Fig. 3a). GR47a probably forms functional strychnine receptor complexes with other unknown GRs.

- The authors should discuss plainly that the 3 Grs expressed in cells expressing Gr64f are not expressed in an "empty" cell, and that these Grs could "pair" also with the other Grs present in these cells (ie Gr64f, Gr5a, Gr61a).

To exclude the possibility that bitter responses in sweet GRNs misexpressing bitter *Grs* were due to abnormal interactions between endogenous sweet *Grs* and ectopically expressed bitter *Grs*, we expressed bitter *Grs* in *Gr64* cluster-deleted sweet GRNs and measured bitter chemical-evoked action potentials. As expected, *Gr64* cluster deletion (*Gr64af*) abolishes sucrose-evoked nerve firings. When we expressed *Gr32a*, *Gr66a*, and *Gr59c* together in the sweet GRNs of *Gr64af*, we still observed responses to LOB, BER, and DEN. Also, when we expressed the *Gr32a*, *Gr66a*, and *Gr22e* combination in the sweet GRNs of *Gr64af*, we observed responses to STR as well as LOB, BER, DEN. Unfortunately, we could not test completely sugar blind flies because of the difficulty of the required genetic crosses. The sweet GRNs of *Gr64af* may express *Gr5a* and *Gr61a*. We could not completely exclude the possibility of interactions between GR5a/GR61a and ectopically expressed bitter GRs, but these are highly unlikely. These results are consistent with our hypothesis that GR32a/GR66a/GR59c and GR32a/GR66a/GR22e can form functional receptors for LOB/BER/DEN and LOB/BER/DEN/STR, respectively. We added these results to Supplementary

Figures 1c and 3a and explained them in the text. Please see page 7 and page 9.

The *Gr64af* flies harboring the deletion of the Gr64 cluster coding region will be described in detail in an independent manuscript.

In addition to these general remarks, here are a few suggestions:

- It would be interesting to provide a table of the constructions used. For example, the naming of the construction used in figure 1 is a bit obscure. I would be interested to know why 2 *Gr64f* Gal4 drivers are noted, on which chromosomes they are inserted and also if the authors have controlled the level of expression of these Grs by RT-PCR for example. Is the level of expression of 1 UAS-X driven by 1 Gal4 equivalent the level of expression of 5-6 UAS-X driven by 1 or 2 gal4?

As suggested, we added the genotypes of each group of flies to the new Supplementary Table 6.

We also changed Figure 1e to increase readability.

We conducted a qPCR analysis to measure the expression level of each *Gr* in three different genotypes: *Gr64f-GAL4*, *Gr64f-GAL4>6 UAS-Grs*, and *2X Gr64f-GAL4>6 UAS-Grs*. We found all six *Grs* are expressed at reasonable levels in sweet GRNs. GAL4 transgene copy number does not affect the expression level of each *Gr*. In addition, LOB, BER, and DEN responses also do not depend on *GAL4* transgene copy number (Please see below). Now, we have rewritten the main text on page 6 to indicate that we "... confirmed the misexpression of each the *Gr* in the sweet GRNs by quantitative PCR (Supplementary Table 1)."

- Figure 1f & figure 3 f: did you test the response to sucrose? Is it affected by your constructions? Did you test the other parent, ie *Gr64f-Gal4*?

We confirmed misexpression of bitter *Grs* in the sweet GRNs did not alter sucrose response. We added these results in Figure 1e, 1f, and 3f and explained the results in main text. Please see page 7 and page 9. We also tested *Gr64f-GAL4* parent line and did not find any difference with either control or UAS parent line (Please see below).

- Figure 2 could be paired with figure 3e.

We prefer keeping these figures separate. We feel it makes it easier to understand the manuscript.

- The electrophysiological recordings (figure 3d, figure 2a LOB) show nice large spikes but also a fairly strong activity of another taste cell. What is this activity? Did you observe this repeatedly?

As suggested, we replaced the representative traces in Figs. 2a and 3d.

- Figure 4: Did you test STR with Gr59c1 constructions? Could you make this figure more explicit? In its present form, it looks quite confusing.

We conducted the suggested experiment and added the result to Figure 4h.

- Figure 5: the statistical tests were done by comparing the responses of the single or double mutants to that of the wild-type. However, it looks that for some situations, the responses of the double-mutants are higher than those of the single mutants. Are these differences statistically relevant? This is the case for : S-a: DEN, S-b: LOB, S-b: DEN. One would expect the double mutant to respond at least at the lowest level of the 2 single mutants.

Now, we show the results of the statistical analysis more explicitly in Figure 5 and discuss some possible reasons for the result in the Discussion.

- Suppl 4: the lower part is mislabeled Gr32a, Gr66a, Gr22e instead of Gr32a, Gr66a, Gr59c.

We made the indicated correction.

We hope you agree that our revised manuscript is now much stronger and that it is now ready for publication in Nature Communications.

Reviewers' comments:

Reviewer #1 (Remarks to the Author):

The authors have addressed all my concerns successfully. I do not have additional comments.

Reviewer #3 (Remarks to the Author):

The authors have answered most of my remarks. The addition of PCR measures is a good move and they have performed additional observations as requested.

However, I have still a few editorial remarks.

line 53 "It remains unclear, however, whether the bitter taste modality perceives a generic "bitterness" leading to a generic behavioral aversion or whether the system shows more specific molecular discernment permitting more subtle, complex behavioral responses."

This issue is not addressed in your paper. You should at least cite a reference as this question has been experimentally addressed in: Masek, P. and Scott, K. (2010). "Limited taste discrimination in *Drosophila*." *Proceedings of the National Academy of Sciences of the United States of America* 107(33): 14833-14838. This is also discussed in several papers from Carlson's group.

line 60 - you mention that taste sensilla are present on the labellum, legs, pharynx and genitalia. You omit sensilla located on the wings. Why? there are as numerous on the wings as on the front legs and they are documented in a number of papers:

the sensilla are described in :

Stocker, R. F. (1994). "The organization of the chemosensory system in *Drosophila melanogaster* - a review." *Cell and Tissue Research* 275(1): 3-26.

Shanbhag, S. R., Park, S. K., Pikielny, C. W. and Steinbrecht, R. A. (2001). "Gustatory organs of *Drosophila melanogaster*: fine structure and expression of the putative odorant-binding protein PBPRP2." *Cell and Tissue Research* 304(3): 423-437.

Yanagawa, A., Guigue, A. M. A. and Marion-Poll, F. (2014). "Hygienic grooming is induced by contact chemicals in *Drosophila melanogaster*." *Frontiers in Behavioral Neuroscience* 8.

Raad, H., Ferveur, J.-F., Ledger, N., Capovilla, M. and Robichon, A. (2016). "Functional gustatory role of chemoreceptors in *Drosophila* wings." *Cell Reports* 15(7): 1442-1454.

Some of their central projections are described in:

Kwon, J. Y., Dahanukar, A., Weiss, L. A. and Carlson, J. R. (2014). "A map of taste neuron projections in the *Drosophila* CNS." *Journal of Biosciences* 39(4): 565-574.

line 83: "In other words, they assumed uniformity among the bitter GRNs." - I would delete

this sentence as this hypothesis was not explicitly expressed in the papers you cite (unless I am mistaken). The main reason of this "simplification" is that electrophysiology observations take time and are difficult to perform on each sensillum. Actually, the heterogeneity of responses you refer to have been demonstrated experimentally in *Drosophila* in Meunier, N., Marion-Poll, F., Rospars, J. P. and Tanimura, T. (2003). "Peripheral coding of bitter taste in *Drosophila*." *Journal of Neurobiology* 56(2): 139-152.

line 247 "a very slight difference" ??

line 360 "Gr64af harboring the deletion of the whole Gr64 cluster will be described in a separate manuscript". The different members of this cluster have already been studied by several groups which have published several mutants of each of the genes of the cluster or of groups of these genes:

- Dahanukar, A., Lei, Y.-T., Kwon, J. Y. and Carlson, J. R. (2007). "Two Gr genes underlie sugar reception in *Drosophila*." *Neuron* 56(3): 503-516.
- Jiao, Y., Moon, S. J. and Montell, C. (2007). "A *Drosophila* gustatory receptor required for the responses to sucrose, glucose, and maltose identified by mRNA tagging." *Proceedings of the National Academy of Sciences of the United States of America* 104(35): 14110-14115.
- Slone, J., Daniels, J. and Amrein, H. (2007). "Sugar receptors in *Drosophila*." *Current Biology* 17(20): 1809-1816.
- Jiao, Y. C., Moon, S. J., Wang, X. Y., Ren, Q. T. and Montell, C. (2008). "Gr64f is required in combination with other gustatory receptors for sugar detection in *Drosophila*." *Current Biology* 18(22): 1797-1801.

and more recently:

- Yavuz, A., Jagge, C., Slone, J. and Amrein, H. (2014). "A genetic tool kit for cellular and behavioral analyses of insect sugar receptors." *Fly* 8(4): 189-196.

What is the rationale of deleting the entire gene cluster including the exons? Furthermore, I find difficult to accept to publish results obtained with a genetic construction which is undocumented in the manuscript where it is used.

Supplementary data:

- I could not find a legend for table S1.

- There are mistakes in the number of transcripts reported in sweet GRNs. If you subtract the values listed in the section "total number of transcripts", the differences are now:

Gr32a	Gr33a	Gr39a.a	Gr59c	Gr66a	Gr89a	
1xGr64f>6Gr6	*5.90E+09	3.10E+09	7.50E+08	1.62E+09	1.10E+09	6.4 1E+09
2xGr64f>6Gr6	*4.20E+09	1.70E+09	9.50E+08	2.02E+09	*6.00E+08	8. 41E+09

It looks as if values were inverted in column Gr32a (either in the calculated values or the original) and Gr66a - 2x is slightly different 5.5 instead of 6.

Rebuttal letter:

- The electrophysiological recordings (figure 3d, figure 2a LOB) show nice large spikes but also a fairly strong activity of another taste cell. What is this activity? Did you observe this repeatedly?

As suggested, we replaced the representative traces in Figs. 2a and 3d.

You did not answer my question - how often did you observe such activity? were these "small" spikes included in the analysis or discarded?

Response to Reviewer #3:

line 53 "It remains unclear, however, whether the bitter taste modality perceives a generic "bitterness" leading to a generic behavioral aversion or whether the system shows more specific molecular discernment permitting more subtle, complex behavioral responses."

This issue is not addressed in your paper. You should at least cite a reference as this question has been experimentally addressed in: Masek, P. and Scott, K. (2010). "Limited taste discrimination in *Drosophila*." *Proceedings of the National Academy of Sciences of the United States of America* 107(33): 14833-14838. This is also discussed in several papers from Carlson's group.

We added the suggested reference in line 53.

line 60 - you mention that taste sensilla are present on the labellum, legs, pharynx and genitalia. You omit sensilla located on the wings. Why? there are as numerous on the wings as on the front legs and they are documented in a number of papers:

the sensilla are described in :

Stocker, R. F. (1994). "The organization of the chemosensory system in *Drosophila melanogaster* - a review." *Cell and Tissue Research* 275(1): 3-26.

Shanbhag, S. R., Park, S. K., Pikielny, C. W. and Steinbrecht, R. A. (2001). "Gustatory organs of *Drosophila melanogaster*: fine structure and expression of the putative odorant-binding protein PBPRP2." *Cell and Tissue Research* 304(3): 423-437.

Yanagawa, A., Guigue, A. M. A. and Marion-Poll, F. (2014). "Hygienic grooming is induced by contact chemicals in *Drosophila melanogaster*." *Frontiers in Behavioral Neuroscience* 8.

Raad, H., Ferveur, J.-F., Ledger, N., Capovilla, M. and Robichon, A. (2016). "Functional gustatory role of chemoreceptors in *Drosophila* wings." *Cell Reports* 15(7): 1442-1454.

Some of their central projections are described in:

Kwon, J. Y., Dahanukar, A., Weiss, L. A. and Carlson, J. R. (2014). "A map of taste neuron projections in the *Drosophila* CNS." *Journal of Biosciences* 39(4): 565-574.

Since the relevant references were already cited, we added a mention of the GRNs in sensilla on the anterior wing margin to line 61.

line 83: "In other words, they assumed uniformity among the bitter GRNs." - I would delete this sentence as this hypothesis was not explicitly expressed in the papers you cite (unless I am mistaken). The main reason of this "simplification" is that electrophysiology observations take time and are difficult to perform on each sensillum. Actually, the heterogeneity of responses you refer to have been demonstrated experimentally in *Drosophila* in Meunier, N., Marion-Poll, F., Rospars, J. P. and Tanimura, T. (2003). "Peripheral coding of bitter taste in *Drosophila*." *Journal of Neurobiology* 56(2): 139-152.

We have removed the offending sentence.

line 247 "a very slight difference" ??

We changed the corresponding sentence as below (page 12 line 252):

"...LOB, BER, and DEN, we found the S-a sensilla in the *Gr22e,Gr59c* double mutant does not show a significant reduction in response to LOB, BER, and DEN compared to the wild-type or single mutant with the exception of the comparison between the *Gr22e,Gr59c* double mutant and the *Gr59c* mutant in response to LOB (Fig. 5a)."

line 360 "Gr64af harboring the deletion of the whole Gr64 cluster will be described in a separate manuscript". The different members of this cluster have already been studied by several groups which have published several mutants of each of the genes of the

cluster or of groups of these genes:

- Dahanukar, A., Lei, Y.-T., Kwon, J. Y. and Carlson, J. R. (2007). "Two Gr genes underlie sugar reception in *Drosophila*." *Neuron* 56(3): 503-516.
- Jiao, Y., Moon, S. J. and Montell, C. (2007). "A *Drosophila* gustatory receptor required for the responses to sucrose, glucose, and maltose identified by mRNA tagging." *Proceedings of the National Academy of Sciences of the United States of America* 104(35): 14110-14115.
- Slone, J., Daniels, J. and Amrein, H. (2007). "Sugar receptors in *Drosophila*." *Current Biology* 17(20): 1809-1816.
- Jiao, Y. C., Moon, S. J., Wang, X. Y., Ren, Q. T. and Montell, C. (2008). "Gr64f is required in combination with other gustatory receptors for sugar detection in *Drosophila*." *Current Biology* 18(22): 1797-1801.

and more recently:

- Yavuz, A., Jagge, C., Slone, J. and Amrein, H. (2014). "A genetic tool kit for cellular and behavioral analyses of insect sugar receptors." *Fly* 8(4): 189-196.

What is the rationale of deleting the entire gene cluster including the exons?

Furthermore, I find difficult to accept to publish results obtained with a genetic construction which is undocumented in the manuscript where it is used.

The *Gr64* cluster consists of six tandem *Gr* genes (*Gr64a-Gr64f*). Expression of bitter GRs in the sweet GRNs of a *Gr64* cluster deletion mutant is a useful control experiment to exclude the possibility of erratic interactions between endogenous sweet GRs and ectopically expressed bitter GRs. The previously reported *Gr64* cluster deletion ($\Delta Gr64$) is lethal due to the additional deletion of neighboring genes. Due to the difficulty of obtaining the desired genotype via genetic crosses using $\Delta Gr64$, we used our new *Gr64* deletion allele, which will be described in a separate manuscript soon (see below figure).

Since generation of the *Gr64* deletion allele is beyond the scope of this manuscript, we would like to leave out the detailed characterization of the *Gr64* deletion allele. We have added to the main text on page 7 a brief description of the rationale for generating a new full deletion of the *Gr64* cluster and a description of how it was constructed. The text now reads, Furthermore, misexpression of *Gr32a*, *Gr59c*, and *Gr66a* in the sweet GRNs of a new *Gr64* cluster deletion mutant (*Gr64af*) covering the coding regions of the entire *Gr64* cluster conferred a similar level of sensitivity to LOB, BER, and DEN as that in wild-type sweet GRNs (Supplementary Fig. 1c). Previous mutants of the *Gr64* cluster either deleted additional neighboring genes or only deleted some genes of the cluster, and were thus less suitable for this purpose. Loss of six of the nine sweet clade *Grs* does not affect ectopic responses to LOB, BER, and DEN upon *Gr32a*, *Gr59c*, and *Gr66a* expression in sweet GRNs. This indicates that the ectopic responses to LOB, BER, and DEN we observed are not due to erratic interactions between endogenous sweet clade GRs and misexpressed bitter GRs.

Supplementary data:

- I could not find a legend for table S1.

We added a legend for Supplementary Table S1.

- There are mistakes in the number of transcripts reported in sweet GRNs. If you

subtract the values listed in the section " total number of transcripts", the differences are now:

Gr32a Gr33a Gr39a.a Gr59c Gr66a Gr89a

1xGr64f>6Grs *5.90E+09 3.10E+09 7.50E+08 1.62E+09 1.10E+09 6.41E+09

2xGr64f>6Grs *4.20E+09 1.70E+09 9.50E+08 2.02E+09 *6.00E+08 8.41E+09

It looks as if values were inverted in column Gr32a (either in the calculated values or the original) and Gr66a - 2x is slightly different 5.5 instead of 6.

We made a mistake here. We appreciate the reviewer catching this error before publication. We have now fixed the error, especially in the column of *Gr32a*, which was highlighted in yellow.

Regarding the numbers, we calculated the expression level for the ectopically expressed *Grs* by subtracting the expression level of each *Gr* in wild-type flies from the expression of the same *Gr* in flies ectopically expressing bitter *Grs*, rounding to two decimal places to make them easier to read. Thus, the discrepancy for Gr66a-2x appears as a resulting of this rounding. Please see the table below, which displays the actual numbers with their corresponding significant figures.

	Genotype	Genes					
		Gr32a	Gr33a	Gr39a.a	Gr59c	Gr66a	Gr89a
number of transcripts	Gr64f-GAL4	3.8E+09	2.1E+09	2.5E+08	1.8E+08	1.3E+09	2.9E+08
	Gr64f>6 Grs	9.7E+09	5.2E+09	1.0E+09	1.8E+09	2.4E+09	6.7E+09
	Gr64f>6 Grs	8.0E+09	3.8E+09	1.2E+09	2.2E+09	1.9E+09	8.7E+09
Number of transcripts in sweet GRNs	Gr64f-GAL4	—	—	—	—	—	—
	Gr64f>6 Grs	5.9E+09	3.1E+09	7.8E+08	1.6E+09	1.1E+09	6.4E+09
	2x Gr64f>6 Grs	4.2E+09	1.7E+09	9.2E+08	2.0E+09	5.5E+08	8.4E+09

Table showing significant figures. We did not include SEM for simplicity.

	Genotype	Genes					
		Gr32a	Gr33a	Gr39a.a	Gr59c	Gr66a	Gr89a
Total number of transcripts	Gr64f-GAL4	3,797,525,922	2,086,257,879	245,421,971	179,560,280	1,334,634,154	293,535,166
	Gr64f>6 Grs	9,715,803,459	5,156,445,637	1,025,874,627	1,762,041,682	2,388,384,319	6,700,143,190
	Gr64f>6 Grs	7,979,334,799	3,768,929,455	1,169,490,625	2,200,563,030	1,885,892,108	8,671,733,184
Number of transcripts in sweet GRNs	Gr64f-GAL4	—	—	—	—	—	—
	Gr64f>6 Grs	5,918,277,537	3,070,187,757	780,452,656	1,582,481,402	1,053,750,164	6,406,608,023
	2x Gr64f>6 Grs	4,181,808,877	1,682,671,575	924,068,654	2,021,002,750	551,257,953	8,378,198,017

Table showing actually numbers. We did not include SEM for simplicity.

Rebuttal letter:

- The electrophysiological recordings (figure 3d, figure 2a LOB) show nice large spikes but also a fairly strong activity of another taste cell. What is this activity? Did you observe this repeatedly?

As suggested, we replaced the representative traces in Figs. 2a and 3d.

You did not answer my question - how often did you observe such activity? were these "small" spikes included in the analysis or discarded?

Because we do not think the small spikes represent gustatory receptor neurons, we did not include them in the analysis. The small spikes were observed occasionally depending on recording conditions such as the angle between the sensillum and the recording electrode or the movement of the sensillum during recording. Indeed, as labellar mechanosensory neurons are known to produce smaller spikes than gustatory receptor neurons (Sánchez-Alcañiz et al, 2016), we assume these small spikes in our traces represent either the firing of mechanosensory neurons or simple noise.

REVIEWERS' COMMENTS:

Reviewer #3 (Remarks to the Author):

The authors have answered all my questions/suggestions and I agree with their corrections.